# Mitochondria-specific drug release and reactive oxygen species burst induced by polyprodrug nanoreactors can enhance chemotherapy

Wenjia Zhang[1,2], Xianglong Hu [1,2], Qi Shen[1,2] & Da Xing[1,2]

Cancer cells exhibit slightly elevated levels of reactive oxygen species (ROS) compared with normal cells, and approximately 90% of intracellular ROS is produced in mitochondria. In situ mitochondrial ROS amplification is a promising strategy to enhance cancer therapy. Here we report cancer cell and mitochondria dual-targeting polyprodrug nanoreactors (DT-PNs) covalently tethered with a high content of repeating camptothecin (CPT) units, which release initial free CPT in the presence of endogenous mitochondrial ROS (mtROS). The in situ released CPT acts as a cellular respiration inhibitor, inducing mtROS upregulation, thus achieving subsequent self-circulation of CPT release and mtROS burst. This mtROS amplification endows long-term high oxidative stress to induce cancer cell apoptosis. This current strategy of endogenously activated mtROS amplification for enhanced chemodynamic therapy overcomes the short lifespan and action range of ROS, avoids the penetration limitation of exogenous light in photodynamic therapy, and is promising for theranostics.

---

[1] MOE Key Laboratory of Laser Life Science & Institute of Laser Life Science, South China Normal University, 510631 Guangzhou, China. [2] College of Biophotonics, South China Normal University, 510631 Guangzhou, China. Correspondence and requests for materials should be addressed to X.H. (email: xlhu@scnu.edu.cn) or (email: huxlong@mail.ustc.edu.cn) or to D.X. (email: xingda@scnu.edu.cn)

ROS are reactive chemical species and essential for many biological processes, such as cellular proliferation, differentiation and migration[1]. Most cancer cells constantly overproduce ~10-fold level of ROS compared with normal cells due to the oncogenic stimulation, mitochondrial malfunction and increased metabolic activity of cancers[2], thus various stimuli-responsive drug delivery systems have been exploited based on endogenous ROS to achieve on demand drug delivery at targeted lesion sites[3–5]. However, the short lifetime (<0.1 ms), limited diffusion and action range (10~20 nm), and relatively insufficient intracellular level of endogenous ROS often compromise the therapeutic efficiency[6,7]. An ~90% intracellular ROS is demonstrated to be generated in mitochondria, which are the major source of superoxide radical that is the precursor of most ROS species, while some other ROS species are also inevitable by-products of the respiratory chain in mitochondria[8,9]. Furthermore, excessive amounts of ROS can damage lipids, proteins and DNA[10,11], which is associated with changes of mitochondrial functions. Plenty of evidence suggests that mitochondria play a vital role in cellular energy metabolism and apoptotic cell death[12–14]. Herein, it's promising to in situ generate high dosage ROS in mitochondria, which can damage mitochondria and activate the programmed cell death, potentiating the therapeutic outcome in cancer therapy[15,16].

Mitochondria have been widely exploited as the target due to the distinguishing function and structure between normal cells and cancer cells, such as oxidative stress, the transmembrane potential ($\Delta\psi_m$), differences in metabolic activity and mtDNA sequence[17–20]. Some mitochondria-targeting photodynamic therapy (PDT) systems have been developed to in situ generate ROS in mitochondria, exhibiting remarkable therapeutic potency[21–24], but it is inescapable to face the penetration limitation of exogenous light. Herein, mitochondrial targeting drug delivery systems in responsive to endogenous signals are expected to maximize the efficiency and minimize potential side effects. On the other hand, one polyprodrug strategy has been originally coined and developed since 2013[25], which can covalently tether repeating prodrug units and deliver high-dosage parent drug at lesion sites, possessing flexible design of polymer topologies, self-assembling morphologies, and theranostic functions[26–35]. Furthermore, chemodynamic therapy is an emerging strategy that can use endogenous chemical energy to produce cytotoxic reactive species and induce cell death in the absence of light irradiation, thus circumventing the penetration limitations through tissues in traditional photodynamic processes[16,36,37].

Here we develop an ROS-responsive polyprodrug nanoreactor with cancer cells and mitochondria dual-targeting property, demonstrating self-circulation of mitochondrial drug release and mtROS burst for enhanced cancer chemodynamic therapy. Camptothecin (CPT) is selected as a model mitochondrial drug, which can act as a cellular respiration inhibitor to stimulate endogenous mtROS production and hyperpolarization of mitochondria[38,39], apart from the general inhibition of DNA topoisomerase I for cancer therapy[40]. First, ROS-responsive CPT prodrug monomer with a thioketal linkage, CPTSM, is prepared, then cancer-targeting polyprodrug amphiphiles, cRGD-PDMA-b-PCPTSM, and mitochondrial-targeting drugs, TPP-PDMA-b-PCPTSM are fabricated, including hydrophilic poly(dimethylacrylamide) (PDMA) and hydrophobic polymerized prodrug block of PCPTSM, further decorating with triphenylphosphonium (TPP) or cyclic Arg-Gly-Asp (cRGD) peptide, respectively. Dual-targeted polyprodrug nanoparticles (DT-PNs) are fabricated from their aqueous co-assembly (Fig. 1). Upon active targeting to mitochondria, endogenous upregulated mtROS in cancer cells can induce initial free CPT release in mitochondria. In situ released CPT further triggers the circulatng increase of mtROS,

achieving subsequent amplification of high-dosage CPT release and a final mtROS burst, which are favorable for long-term high oxidative stress to efficiently eliminate cancer cells. Quantitative evaluation of ROS level is performed at multiple levels, including in situ mitochondrial superoxide, intracellular total ROS and intratumor ROS inside tumor-bearing mice.

## Results

**Synthesis of ROS-responsive prodrug monomer and polyprodrug.** CPT is a cytotoxic quinoline alkaloid isolated from the Happy Tree, and is frequently employed for cancer treatment in traditional Chinese medicine. We chose the extremely hydrophobic CPT as a model drug in virtue of its anticancer activity and facile functionalization, and most importantly, CPT is also a cellular respiration inhibitor that can induce excessive ROS generation in mitochondria[38]. Here, a ROS-responsive CPT prodrug monomer, CPTSM, was synthesized by introducing a ROS-cleavable thioketal functional moiety to covalently modify the hydroxyl group in the CPT parent drug (Supplementary Fig. 1)[30]. The chemical structure of CPTSM was verified by [1]H NMR and [13]C NMR spectra (Supplementary Fig. 2 and 3). ROS-responsive polyprodrug amphiphiles with distinct targeting moieties, cell targeting cRGD-PDMA-b-PCPTSM and mitochondria targeting TPP-PDMA-b-PCPTSM, were synthesized via two-step reversible addition fragmentation transfer (RAFT) polymerization and verified by [1]H NMR spectra (Supplementary Fig. 4 and 5)[41]. A typical absorption peak of CPT at ~370 nm was observed in the absorbance spectra of these ROS-responsive polyprodrug amphiphiles (Fig. 2a). Detailed structural parameters were summarized (Supplementary Table 1), in which the CPT loading content was >27 wt% for these ROS-responsive polyprodrug amphiphiles. It can be envisaged that the repeating ROS-cleavable CPT prodrug units within polyprodrug amphiphiles not only realize high drug loading and formulation stability, but also provide the capability to release high dosage CPT parent drug in response to oxidative milieu in mitochondria (Fig. 1).

**Fabrication of dual-targeted polyprodrug nanoreactors.** For cancer cell imaging and intracellular trafficking, cancer cell targeting cRGD-PDMA-b-P(CPTSM-co-RhB) was labeled with a small amount of RhB via the copolymerization of RhB monomer with CPTSM in the hydrophobic block[42]. The aqueous self-assembly of cRGD-PDMA-b-P(CPTSM-co-RhB) and mitochondria targeting TPP-PDMA-b-PCPTSM could fabricate dual-targeted polyprodrug nanoparticles (denoted as DT-PNs) in aqueous media, exhibiting relatively high CPT loading content, up to ~30.8 wt% (Fig. 1). Thereby, an ROS-responsive PCPTSM block was located in the hydrophobic core, and the hydrophilic corona was decorated with cRGD and TPP targeting species. The UV/Vis absorption spectrum of RhB-labelled DT-PNs dispersion displayed typical signals of RhB at ~580 nm and the peak of CPT at ~370 nm (Fig. 2a). Furthermore, dynamic light scattering (DLS) analysis revealed that the average diameter of DT-PNs was ~55 nm, the uniform spherical morphology was confirmed by transmission electron microscopy (TEM) analysis (Fig. 2b). The stability of DT-PNs was examined in water, PBS, DMEM culture medium with 10 % fetal bovine serum (FBS), at 37 °C. No obvious size change was observed over 7 days, indicating the stability of DT-PNs (Supplementary Fig. 6).

**In vitro drug release at various simulated ROS levels.** For the polyprodrug design in this work, CPT was covalently tethered by a thioketal linker in the side chain of polyprodrug amphiphiles, exhibiting potential ROS-responsive drug release. To investigate the ROS-induced CPT release, $KO_2$ dissolved into dry DMSO was

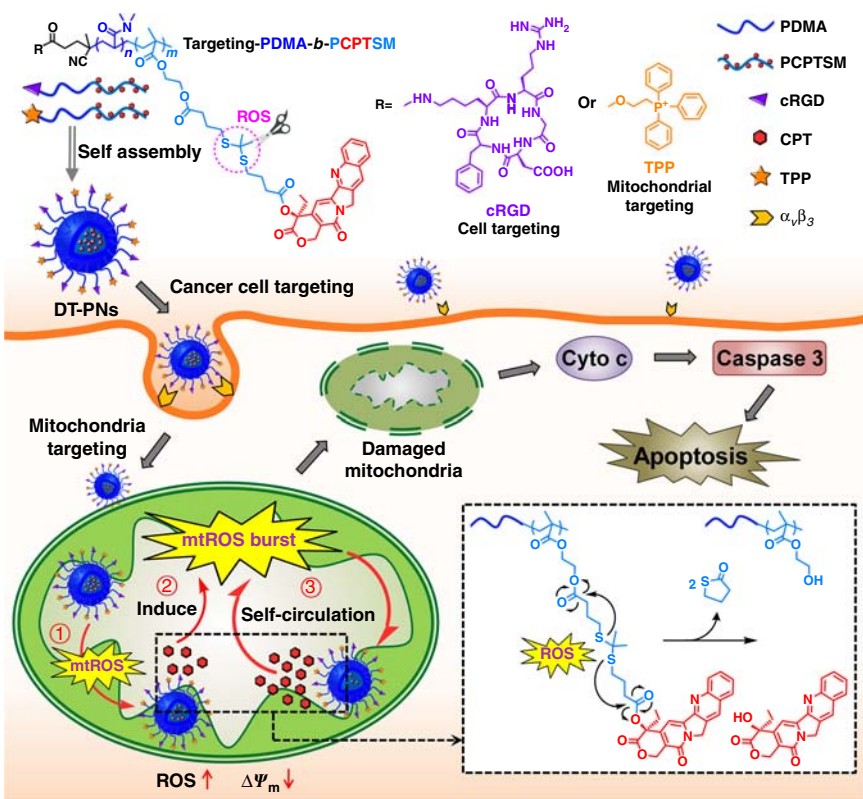

**Fig. 1** Mitochondria-specific polyprodrug for self-circulation drug release with ROS burst. Dual-targeting polyprodug nanoreactors (DT-PNs) were fabricated from the co-assembly of cancer targeting polyprodrug amphiphiles, cRGD-PDMA-*b*-PCPTSM and mitochondria targeting ones, TPP-PDMA-*b*-PCPTSM. Upon cellular uptake by cancer cells, DT-PNs can specifically enter mitochondria, then low-level endogenous mitochondrial ROS (mtROS) can induce initial slight CPT release, which stimulates significant mtROS regeneration, thus further releasing much more CPT. Finally, the self-circulation of CPT Release and ROS burst results in highly oxidative stress, thus initiating mitochondrial dysfunction and cell apoptosis for enhanced cancer chemodynamic therapy

employed to generate superoxide $(O_2^-)$[5,43], The typical Fenton reaction between $Fe^{2+}$ and $H_2O_2$ was used to generate hydroxyl radical $(\cdot OH)$[44], sodium hypochlorite (NaOCl) at pH 6.02 to form hypochlorous acid (HClO), $C_{ClO^-} = Abs_{292nm}/0.39$ (mM)[45], $ONOO^-$ was prepared according to the reported method, $C_{ONOO^-} = Abs_{302nm}/1.67$ (mM)[46]. HPLC analysis showed that PDMA-*b*-PCPTSM could readily release CPT in the presence of five types of ROS at 1 mM, including $O_2^-$, $H_2O_2$, $\cdot OH$, $ClO^-$ and $ONOO^-$ upon 24 h incubation, respectively (Fig. 2c). The in vitro CPT release rate was further evaluated for DT-PNs upon treating with these five types of ROS (Fig. 2d). $ONOO^-$ and $ClO^-$ were observed to mediate much faster CPT release compared with $O_2^-$, $H_2O_2$, and $\cdot OH$, which also agreed with their relative oxidation potency[4,47].

Additionally, DLS was employed to monitor the degradation profile of DT-PNs at different $H_2O_2$ levels (Fig. 2e). At 1 μM $H_2O_2$, a comparable upper level of $H_2O_2$ in physiological condition and blood circulation, the relative light scattering intensities remained almost constant without detectable degradation after 10 h. At an elevated level of $H_2O_2$ at 0.1 mM, which was similar to the ROS level in cancer cells, less than 20% decrease of light scattering intensities was observed after 10 h incubation, indicating a slight degradation of the DT-PNs in a simulated microenvironment of cancer cells. However, DT-PNs could be degraded at higher $H_2O_2$ levels, decreasing up to ~60% and ~96% at 1 mM and 10 mM $H_2O_2$, respectively. The in vitro CPT release profile was further quantitatively evaluated. The drug leakage was minimal at 1 μM $H_2O_2$, whereas at 0.1 mM $H_2O_2$, there was a ~28% cumulative CPT release upon incubation for 80 h, which was not significant at the simulated ROS level in general

cancer cells. Whereas at 1 mM $H_2O_2$ and 10 mM $H_2O_2$, ~51% and ~90% total release was observed upon treating for 80 h, respectively (Fig. 2f). In vitro drug release analysis with $H_2O_2$ as the trigger suggested that a small amount of CPT can be released from ROS-responsive DT-PNs in cancer cells, but higher ROS level would release a greater amount of CPT. Thioketal linkages exhibited the most efficient degradation in response to the hydroxyl radical[48], thus the exact drug release rates could be promoted in the presence of hydroxyl radical. Collectively, intracellular ROS can release a small amount of the CPT parent drug, whereas at simulated physiological conditions (~1 μM), CPT is hardly released and activated, thus avoiding the potential side effects suffered from drug leakage.

We monitored the ROS-responsive degradation of DT-PNs by Transmission Electron Microscope (TEM). For the control group, the diameter of DT-PNs remained at ~29.3 ± 5.4 nm after 24 h incubation in PBS at pH 7.4, 37 °C. Whereas the particle size of DT-PNs decreased readily after 24 h incubation with five types of ROS at 1 mM concentration; specifically, $O_2^-$ (9.8 ± 2.6 nm), $H_2O_2$ (9.2 ± 2.6 nm), $\cdot OH$ (8.6 ± 1.9 nm), $ClO^-$ (3.9 ± 1.0 nm) and $ONOO^-$ (3.0 ± 1.2 nm), respectively (Fig. 2g). The degradation extent for the groups of $ONOO^-$ and $ClO^-$ was more significant than other groups, which was expected to promote a much faster drug release. Hence, these five kinds of ROS could mediate the cleavage of thioketal linkages and the polyprodrug degradation to release parent CPT drug.

**Dual targeting potency of DT-PNs.** Integrin $\alpha_v\beta_3$ plays a pivotal role in the early stage of tumor angiogenesis and also acts as a receptor for extracellular matrix proteins upon exposure to cRGD[49].

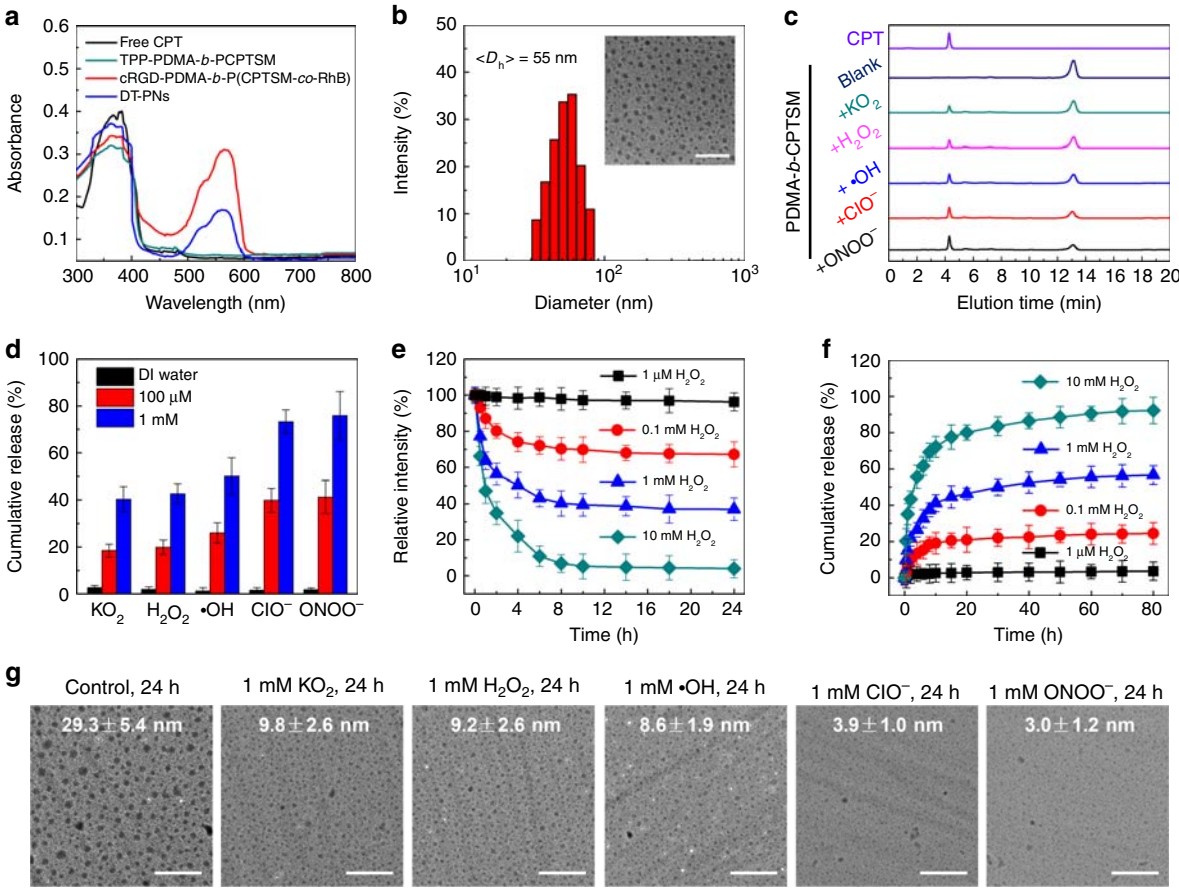

**Fig. 2** Physiochemical characterization and ROS-responsive degradation and drug release. **a** Absorption spectra recorded for free CPT, TPP-PDMA-*b*-PCPTSM, RhB-labelled cRGD-PDMA-*b*-P(CPTSM-*co*-RhB), and the resulting DT-PNs fabricated from cRGD-PDMA-*b*-P(CPTSM-*co*-RhB) and TPP-PDMA-*b*-PCPTSM. **b** Hydrodynamic diameter distribution and TEM image (inset) obtained for DT-PNs (scale bar, 200 nm). **c** HPLC analysis of PDMA-*b*-PCPTSM upon 24 h incubation with five kinds of distinct ROS at 1 mM, including potassium superoxide ($KO_2$), $H_2O_2$, hydroxyl radical (•OH), hypochlorite ($ClO^-$) and peroxynitrite ($ONOO^-$), respectively. The mobile phase was 50/50 acetonitrile and water at a flow rate of 1.0 mL/min. **d** In vitro CPT release from DT-PNs against five kinds of ROS for 24 h with diverse contents. **e** Degradation kinetics of DT-PNs, determined by the normalized scattered light intensities from DLS analysis. **f** In vitro CPT release from DT-PNs against different level of $H_2O_2$ at pH 7.4, 37 °C. **g** TEM images of DT-PNs after 24 h incubation with five types of ROS types at 1 mM, 37 °C, scale bar, 200 nm. The content of DT-PNs (1 mg in 0.5 mL, 0.82 μmol of thioketal groups) was employed in **c-g**

To investigate the cancer targeting property of DT-PNs based on the decoration of cRGD moieties, cellular uptake of RhB-labelled DT-PNs towards two kinds of cancer cells, MCF-7 cells with lower expressed $\alpha_v\beta_3$ integrin and 4T1 cells with overexpressed $\alpha_v\beta_3$ integrin, was compared. We performed pre-blocking experiments based on CLSM imaging and statistical flow cytometry analysis (Fig. 3a, b, and c). Cells were pretreated with excess free cRGD before incubation with DT-PNs[50]. The fluorescence intensity of 4T1 cells was observed to be much stronger than that of MCF-7 cells, or 4T1 cells pre-blocking by free cRGD. Thus, the cRGD moieties in DT-PNs can facilitate high affinity to integrin $\alpha_v\beta_3$-rich cancer cells. Furthermore, flow cytometry analysis was employed to quantitatively examine the tumor targeting efficiency of DT-PNs compared with other polyprodrug nanoreactors (Fig. 3d, e). Four kinds of RhB-labelled polyprodrug nanoreactors were compared, including non-targeting polyprodrug nanoreactors (NT-PNs) formulated from PDMA-*b*-PCPTSM and PDMA-*b*-P(CPTSM-*co*-RhB), mitochondria-targeting polyprodrug nanoreactors TPP-PNs from the co-assembly of TPP-PDMA-*b*-PCPTSM and PDMA-*b*-P(CPTSM-*co*-RhB), cell-targeting polyprodrug nanoreactors cRGD-PNs from the co-assembly of cRGD-PDMA-*b*-P(CPTSM-*co*-RhB) and PDMA-*b*-PCPTSM, as well as the final dual-targeted polyprodrug nanoreactors DT-PNs from the co-assembly of cRGD-

PDMA-*b*-P(CPTSM-*co*-RhB) and TPP-PDMA-*b*-PCPTSM. Although the presence of mitochondria-targeting TPP endowed TPP-PNs with enhanced positive charge compared with NT-PNs (Supplementary Fig. 7), TPP and cRGD dual-formulated DT-PNs demonstrated the highest fluorescence intensity, ~2.6-fold value compared with NT-PNs, thus exhibiting the fastest uptake rate among four kinds of nanoparticles (Fig. 3d, e). These results indicated the superiority of dual-targeting property of DT-PNs compared with non-targeting or a single-targeting modality.

In addition, various mitochondrial drug delivery systems were functionalized with TPP, a polar lipophilic cation that enable molecules/nanoparticles to penetrate and accumulate selectively in the phospholipid bilayer of mitochondrion[51–55]. The mitochondrial targeting property of DT-PNs was then examined in 4T1 cells (Supplementary Fig. 8). CLSM imaging revealed that the co-localization rate exhibited a time-dependent cellular uptake, which gradually enhanced as the incubation time increased. Upon incubation for 4 h, the red fluorescence signal from DT-PNs and green signal from Mitotracker matched well and showed yellow fluorescent spots within mitochondria, demonstrating the specific mitochondria targeting ability of DT-PNs. The line-scan profiles also denoted the co-localization of DT-PNs within mitochondrial compartments. Furthermore, similar results were observed for

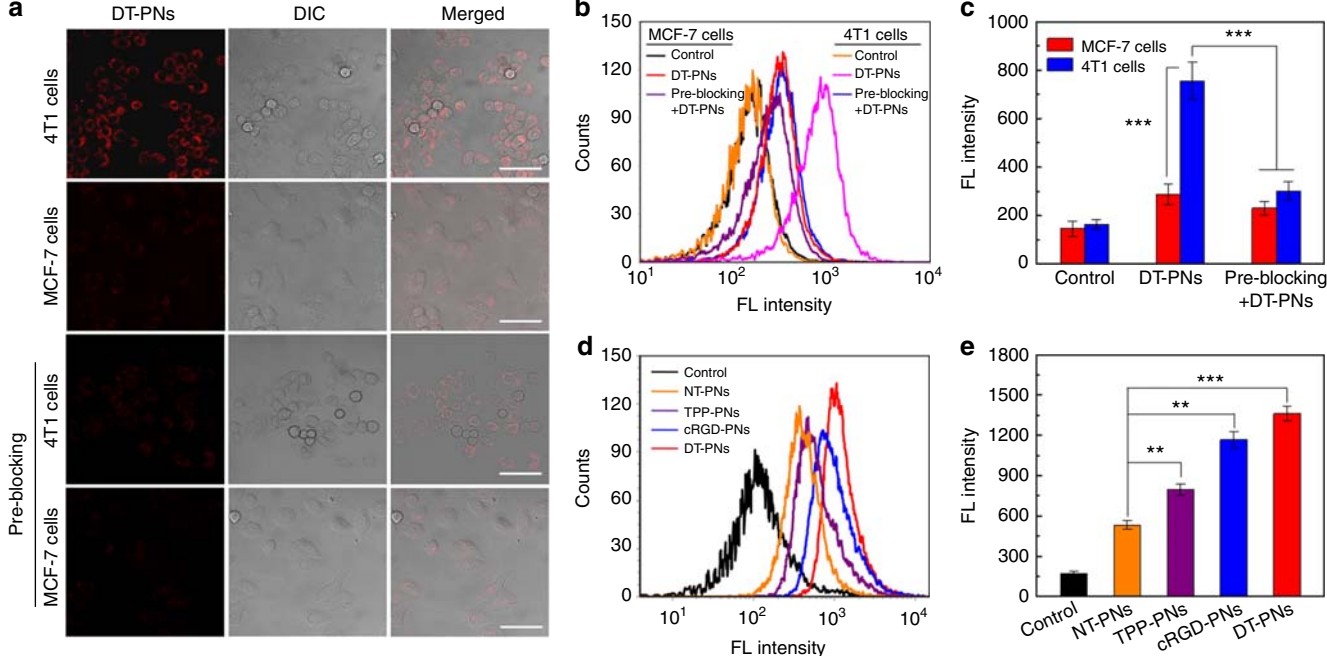

**Fig. 3** Cancer cell targeting property of DT-PNs. **a** CLSM images of 4T1 cells (integrin positive) and MCF-7 cells (integrin negative) after incubating with RhB-labelled DT-PNs under different treatments for 2 h at 37 °C. For the pre-blocking experiment, cells were pretreated with 2 μM free cRGD for 30 min before incubation with DT-PNs (scale bar, 50 μm). **b**, **c** Flow cytometry analysis and the statistical analysis in **a**. The mean value was calculated by the *t* test (mean ± s.e.m. *n* = 3). ***p < 0.001, compared with the indicated group. **d** Flow cytometry analysis of 4T1 cells upon 6 h treatment with RhB-labelled polyprodrug nanoparticles with diverse targeting moieties, including non-targeting nanoparticles (NT-PNs) from the assembly of PDMA-*b*-PCPTSM and PDMA-*b*-P(CPTSM-*co*-RhB); TPP-PNs from the assembly of TPP-PDMA-*b*-PCPTSM and PDMA-*b*-P(CPTSM-*co*-RhB), cRGD-PNs from the assembly of cRGD-PDMA-*b*-P(CPTSM-*co*-RhB) and PDMA-*b*-PCPTSM, and the final resultant DT-PNs. **e** Statistical analysis of the mean fluorescence intensity in **b**. The mean value was calculated by the *t* test (mean ± s.e.m. *n* = 3). **p < 0.01, ***p < 0.001, compared with the indicated group

U87 cells upon incubation with DT-PNs (Supplementary Fig. 9). However, there is little overlap between the fluorescence pixels of DT-PNs and Lysotracker (Supplementary Fig. 10). These results further verified that DT-PNs possessed selectivity for cancer cells' mitochondria.

**In situ mitochondrial drug release and ROS burst**. As shown previously, simulated cancer intracellular ROS can trigger the release of a small amount of CPT (Fig. 2f), together with the specific location of DT-PNs in mitochondria (Supplementary Fig. 8 and 9), in which it can inhibit cellular respiration to potentially induce mtROS increasing and mitochondrial hyperpolarization. To further probe the intracellular fate of CPT drug moieties, 4T1 cells were pretreated with DT-PNs for 2, 4, 8 and 16 h (Fig. 4). CLSM imaging and the line-scan profiles also revealed that DT-PNs were well co-localized with mitochondria. Blue fluorescence ascribing to CPT moieties uniformly dispersed in the mitochondria even after 16 h incubation, thus exhibiting excellent co-localization in mitochondria. Constrained by the small size of mitochondria and the resolution limit of CLSM imaging, the CPT cleavage from the polymer backbone in mitochondria was difficult to observe at the molecular level. Nevertheless, combined with the results as described in Fig. 2c–f, thus intracellular inherent mtROS could also mediate in situ CPT release in mitochondria. More importantly, upon increasing the incubation time, the morphology of mitochondria changed obviously from a normal short tube to a fragmented and punctiform morphology, suggesting potential mitochondrial damage[56]. Thus the mitochondrial fragmentation most probably resulted from the elevated oxidative stress caused by in situ mitochondrial CPT release from DT-PNs.

Subsequently, the ROS state of 4T1 cells was evaluated upon treating with four kinds of polyprodrug nanoparticles with different targeting moieties and free CPT, respectively. The overall intracellular ROS was first evaluated by flow cytometry analysis (FACS) using 2′,7′-dichlorofluorescein diacetate (DCFH-DA), which could be rapidly oxidized by ROS to generate green fluorescent dichlorofluorescein (DCF). Notably, a sharp increase in DCF fluorescence was detected in the DT-PNs group after 8 h incubation. In contrast, the fluorescence intensity change of NT-PNs, cRGD-PNs and TPP-PNs groups was negligible (Supplementary Fig. 11a, b). In addition, the intracellular total ROS up-regulation induced by DT-PNs could be significantly attenuated by two ROS scavengers (20 mM N-acetylcysteine, NAC and 10 mM Vitamin C, Vc)[57,58]. These results indicated that DT-PNs could significantly enhance the intracellular ROS level. Such a ROS burst of DT-PNs was further visualized with confocal laser scanning microscopy (CLSM) imaging (Supplementary Fig. 11c), which was in good agreement with the results shown in Supplementary Fig. 11a, b.

Furthermore, the specific mitochondrial superoxide, the source of most ROS species, was also evaluated upon different treatments. FACS and CLSM imaging were employed to discern the superoxide in mitochondria using MitoSOX Red as an indicator[59]. Real-time kinetic detection of mitochondrial $O_2^-$ by FCAS was performed for living cells upon incubation with diverse samples from 0 to 24 h, respectively (Fig. 5a and Supplementary Fig. 12). At 10 h, the group of DT-PNs reached the highest peak of mitochondrial $O_2^-$ level, ~10.3-fold higher than the control group, which was much higher than other groups with different targeting properties. Interestingly, upon incubation with DT-PNs for 24 h, ~6.8-fold increase in fluorescence intensity was even found, compared to the value at 0 h, suggesting the extended

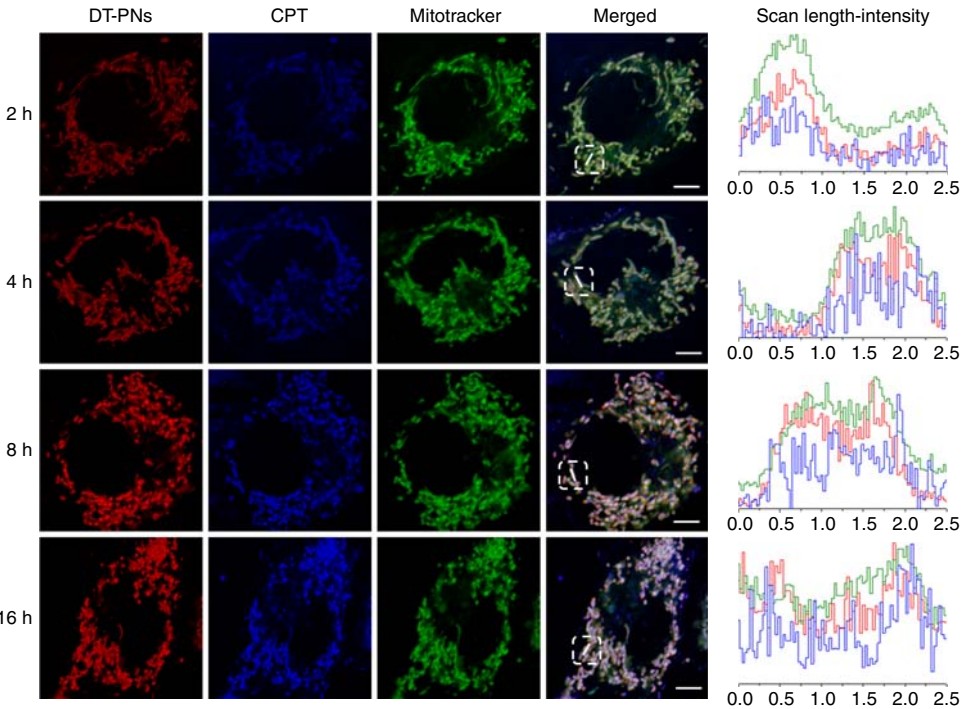

**Fig. 4** Mitochondria-specific localization and in situ drug release in mitochondria. 4T1 cells were treated with RhB-labeled DT-PNs for 2, 4, 8, and 16 h, respectively. RhB (red channel) was utilized to label the polymer backbone. Mitotracker (green channel) was employed to co-stain mitochondria. Blue channel was originated from the emission of CPT itself. In the line-scan profiles, the red, blue, and green curves represent the fluorescence intensity from DT-PNs, CPT, and Mitotracker Green, respectively (scale bar, 5 μm)

oxidative stress exerted by the self-circulation of CPT release and ROS burst. Two ROS scavengers, NAC and Vc, could eliminate ROS in all these groups. Notably, free CPT could also promote moderate elevation of mitochondrial $O_2^-$ at a slightly quick rate, achieving its highest peak at 2 h, ~6.5-fold compared with the control group. Temporarily, fast intracellular diffusion of free CPT with compromised mitochondrial targeting property resulted in the fast upregulation of mitochondrial $O_2^-$, but the enhancing extent and function duration were lower than that of DT-PNs. We evaluated the level of mitochondrial $O_2^-$ by FACS and CLSM imaging for these groups upon 8 h incubation, the highest red fluorescence was observed for the group of DT-PNs. It agreed well with above kinetic monitoring, demonstrating most significant mitochondrial $O_2^-$ generation from DT-PNs, potentially endowed by the in situ mitochondrial CPT release and ROS burst (Fig. 5b, c, d). Herein, the highest level of $O_2^-$ in mitochondria would guarantee the formation of much more amount of ROS species in mitochondria. Thus for cells treated with DT-PNs, the lipophilic and positively charged TPP moiety could efficiently promote the accumulation of DT-PNs in negatively charged mitochondria, in which mtROS was subsequently activated due to the in situ self-promoted CPT release from DT-PNs in mitochondria.

Finally, we further investigated whether the ROS burst induced by DT-PNs could occur in vivo. To this aim, 4T1 tumor-bearing mice were intravenously injected with different formulations for 8 h followed by the intraperitoneal administration of the ROS probe, Cellular Reactive Oxygen Species Detection Assay Kit (Deep Red Fluorescence)[60], and imaged at 1 h, 2 h and 4 h post-administration (Fig. 5e). Interestingly, the fluorescence intensity of tumor sites (white arrow indicated) of free CPT, NT-PNs, cRGD-PNs and TPP-PNs groups exhibited unobvious enhancement at 1 h, 2 h and 4 h. For the evaluation of intracellular total ROS level and specific mitochondrial superoxide, the free CPT group showed moderate enhancement (Fig. 5a–d), but for the

in vivo ROS evaluation, the extent of ROS increasing in tumor sites was comparatively unobvious, potentially due to its poor water solubility, instability and compromised pharmacokinetics[61]. In contrast, for the group treated with DT-PNs, ~12.6-fold fluorescence intensity was detected for DT-PNs at 4 h compared with the control group, suggesting significant ROS level in tumor sites (Fig. 5f). Furthermore, the tumor fluorescence intensity for the DT-PNs group could be significantly attenuated in the presence of ROS scavengers, NAC and Vc. Thus, we envisaged that efficient tumor accumulation of DT-PNs and subsequent mitochondria-specific ROS burst contributed to the in vivo tumor ROS increasing.

**In situ mitochondria damage.** Mitochondria are central organelles, to which the damage can directly activate the intrinsic mitochondrial pathway of apoptosis[62–64]. A variety of pivotal events related to mitochondria can result in cell apoptosis, such as release of caspase-activating protein, cytochrome c (Cyto c), loss of mitochondrial membrane potential and depolarization, disruption of electron transport chain, oxidative phosphorylation, ATP production for cell metabolism, and the interaction with pro-apoptotic Bax and anti-apoptotic Bcl-2 family proteins[60,65]. DT-PNs induced mitochondrial dysfunction was first investigated by evaluating the mitochondrial membrane potential using JC-1 staining (Fig. 6a). It has been shown that the JC-1 probe can examine mitochondrial damage. The green monomer of JC-1 could enter the cytoplasm and aggregate in normal mitochondria, with the formation of numerous red J-aggregate, while the fluorescence transition from red to green suggested the loss of membrane potential and thus significant mitochondrial damage. The green fluorescence (damaged mitochondria) of 4T1 cells in four groups including control, NT-PNs, cRGD-PNs, and TPP-PNs was not significant and red fluorescence (normal mitochondria) was obvious after 12 h incubation. In contrast, the

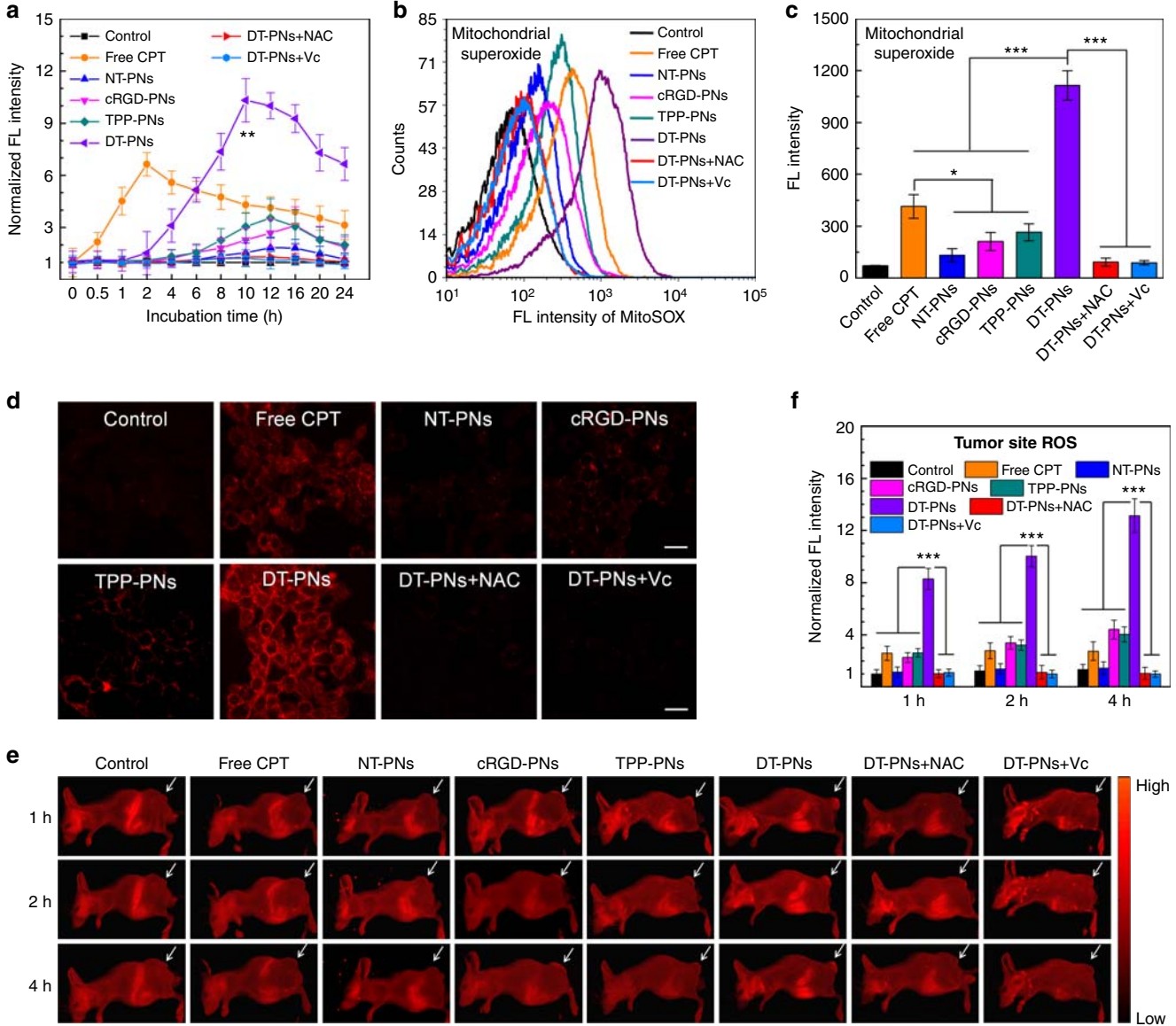

**Fig. 5** In vitro and in vivo demonstration of ROS burst triggered by DT-PNs. **a** Time-dependent relative level of mitochondrial superoxide for 4T1 cells after diverse treatments within 24 h based on flow cytometry analysis (FCAS). MitoSOX Red was employed as a fluorescent indicator of mitochondrial superoxide. The mean value was calculated by the $t$ test (mean ± s.e.m. $n = 3$). **$p < 0.01$, compared with the free CPT group. **b** Determination of superoxide in mitochondria upon different treatments for 8 h detected by MitoSOX based on FCAS. **c** Statistical analysis of the mean fluorescence intensity in **b**. The mean value was calculated by the $t$ test (mean ± s.e.m. $n = 3$). *$p < 0.05$, ***$p < 0.001$, compared with the indicated group. **d** CLSM imaging of mitochondrial superoxide by MitoSOX staining upon the same treatments in **b** (scale bar, 20 µm). **e** In vivo fluorescence imaging of 4T1 tumor-bearing Balb/c mice after intravenous injection with free CPT, NT-PNs, cRGD-PNs, TPP-PNs, and DT-PNs with/without 20 mM NAC or 10 mM Vc for 12 h, and followed by intraperitoneal injection with one ROS probe (Cellular Reactive Oxygen Species Detection Assay Kit, Deep Red Fluorescence), white arrows indicate the tumor sites. **f** Quantitative analysis for the fluorescence intensity of tumor sites in **e**. The mean value was calculated by the $t$ test (mean ± s.e. m. $n = 3$). ***$p < 0.001$, compared with the indicated group

disappearance of red fluorescence and increase of green fluorescence (monomer in the cytoplasm) could be observed in 4T1 cells after treatment with DT-PNs. Notably, free CPT could also induce moderate mitochondrial dysfunction based on the JC-1 staining, which was consistent with previous report[38,66].

Furthermore, to verify the hypothesis that the mitochondrial fragmentation was attributed to the elevated mtROS caused by DT-PNs (Fig. 4), upon different treatments, the morphology of mitochondria was presented based on the staining by commercial MitoTracker Red (Fig. 6b). The mitochondrial morphology in some groups, such as control, free CPT, NT-PNs, cRGD-PNs, TPP-PNs groups, was approximate to the normal tubular network.

Whereas upon treating with DT-PNs for 12 h, the mitochondria were fragmented with a punctiform morphology, and the post-treatment with different antioxidants (NAC and Vc) could prevent the mitochondrial fragmentation to some extent due to the effect of ROS elimination (Fig. 6b). With the localization of DT-PNs in mitochondria and subsequent ROS burst, the mitochondrial fragmentation would exhibit relatively low Δψm.

To further demonstrate the apoptotic pathway mediated by mitochondria, cellular ATP level and typical western blot analysis were performed to examine the apoptosis-related proteins. Upon incubation with diverse samples for 12 or 24 h, cellular ATP production was analyzed, the DT-PNs group exhibited most

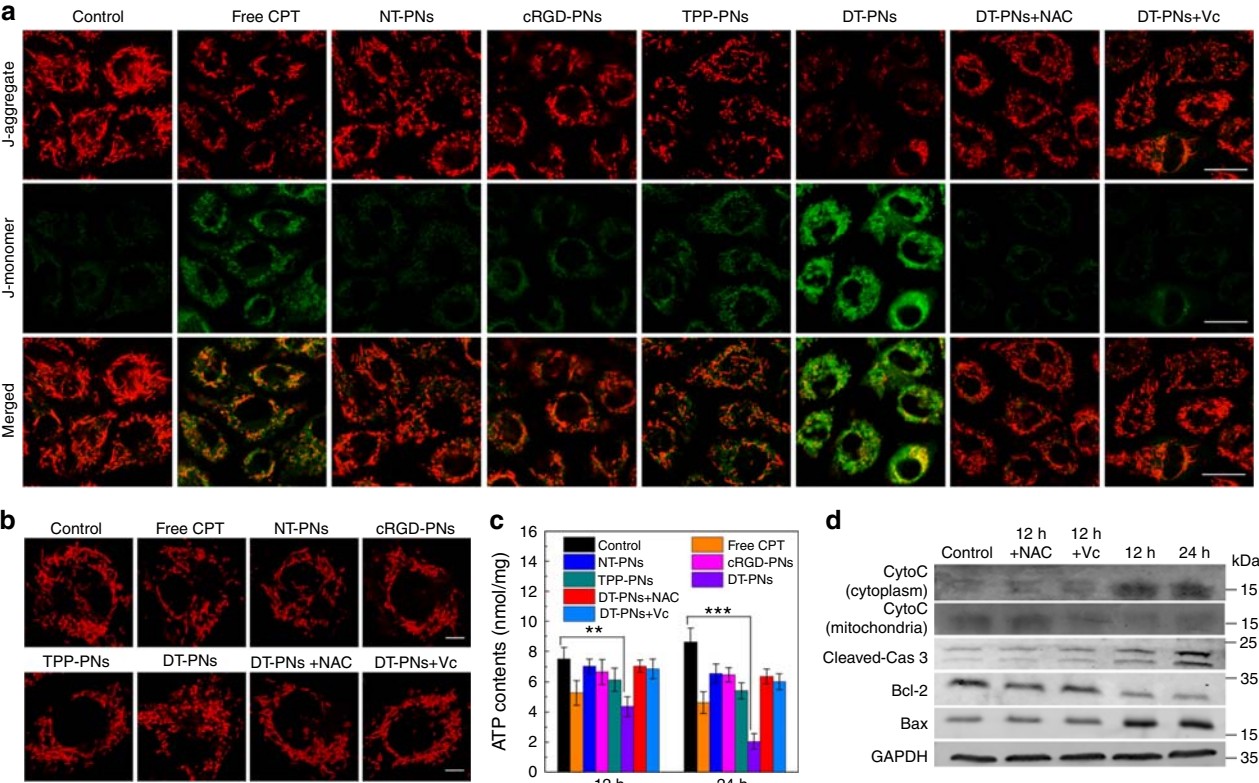

**Fig. 6** Mitochondrial damage-induced apoptosis of cancer cells by DT-PNs. **a** CLSM images of JC-1 stained cells after different treatments, the fluorescence transition from red to green indicated significant mitochondrial damage (scale bar, 20 μm). **b** CLSM imaging of mitochondrial morphology in 4T1 cells after incubation for 24 h with DT-PNs, the cells were stained with MitoTracker Red (scale bar, 5 μm). **c** 4T1 cells were treated with DT-PNs for 12 h and 24 h for the cellular ATP measurement. The mean value was calculated by the $t$ test (mean ± s.e.m. $n = 3$). \*\*$p < 0.01$, \*\*\*$p < 0.001$, compared with the indicated group. **d** Western blotting analysis for the expression of Cyto C, cleaved-caspase 3, Bcl-2, and Bax proteins after treating with DT-PNs for 12 and 24 h or with NAC and Vc prior to treatment, GAPDH was used as a control

significant ATP decrease, up to ~42.2% at 12 h, and ~76.3% at 24 h compared with that of control (Fig. 6c). The apoptosis-related proteins in 4T1 cells were further detected by western blot analysis (Fig. 6d and Supplementary Fig. 13). After incubation with DT-PNs for 12 h or 24 h, the expression of cytochrome c in cytoplasm and caspase-3 both increased dramatically, and the pretreatment with antioxidants (NAC or Vc) would remit the upregulation of cytochrome c and cleaved caspase-3 in cytoplasm. Conversely, the expression of anti-apoptotic Bcl-2 protein was significantly inhibited, and the pro-apoptotic Bax increased greatly. These results verified the mitochondria mediated apoptotic pathway triggered by DT-PNs[39]. Based on the above results, we concluded that the mtROS triggered in situ mitochondrial CPT release could amplify oxidative stress (mtROS) and decrease the mitochondrial membrane potential, which could result in mitochondrial damage to initiate programmed cell death. Therefore, the dual-targeting polyprodrug nanoreactors could act as polyprodrug nanoreactors to endogenously activate in situ mitochondrial drug release and ROS burst, exerting persistent oxidative stress for enhanced cancer chemodynamic therapy.

**In vitro cell viability**. To investigate the in vitro cytotoxicity of dual-targeted polyprodrug nanoreactors, MTT assays were performed with 4T1 cells upon incubation with polyprodrug micelles tethered with different targeting moieties (Fig. 7a). DT-PNs showed comparable cytotoxicity compared with that of free CPT, however, other nanoparticles did not have a marked effect on viability due to their lack of dual targeting property. Similarly, two selected

antioxidants (NAC and Vc) could counteract the cytotoxicity induced by DT-PNs (Supplementary Fig. 14). Additionally, fluorescein-annexin V (V-FITC) and propidium iodide (PI) double staining assays further demonstrated that the cell toxicity of polyprodrug nanoreactors was associated with early apoptosis and late apoptosis. The apoptotic ratio induced by DT-PNs, 45.73%, was significantly higher than that of NT-PNs (10.03%), TPP-PNs (14.87%), cRGD-PNs (19.98%), as well as free CPT (34.27%) in the same condition (Fig. 7b, c). the anticancer effects, which was further confirmed by the Calcein AM/PI double assay (Fig. 7d). The red fluorescent spots indicated dead cells, and the green spots living cells. The highest red/green ratio was observed for DT-PNs, suggesting the most effective inhibition of cancer cells by DT-PNs.

**In vivo biodistribution**. The decoration of cRGD endowed the potential enrichment effect of DT-PNs in cancer cells. ICG-loaded DT-PNs were employed to investigate the biodistribution of DT-PNs based on the ICG fluorescence (Supplementary Fig. 15). The fluorescence of tumor sites was intensely increased at 8 h post-injection with DT-PNs, and still maintained strong signals at 24 h. However, no obvious fluorescence increase was detected at 8 and 24 h post-injection with NT-PNs. Furthermore, for free ICG-treated group, the fluorescence intensity of tumor sites was very limited, and decreased quickly (Fig. 8a). The ex vivo fluorescence images of major organs and tumors were obtained at 24 h, the tumor fluorescence intensity of DT-PNs treated mice was nearly ~4-fold higher than that of free ICG, and ~2-fold than NT-PNs (Fig. 8b, c). Thus DT-PNs exhibited noteworthy tumor

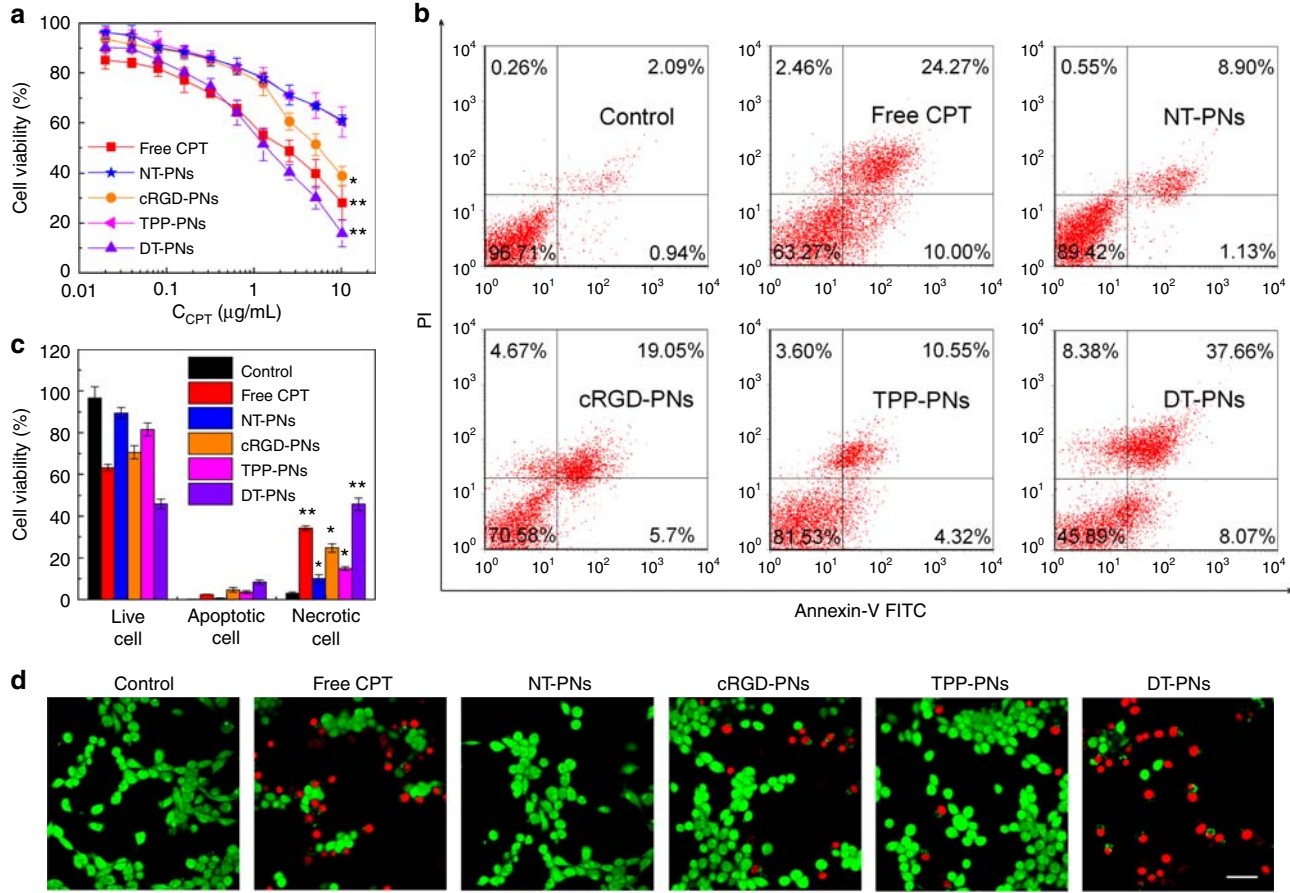

**Fig. 7** In vitro cytotoxicity and the analysis of apoptosis/necrosis. **a** In vitro cytotoxicity determined by MTT assay against 4T1 cells upon 36 h treatment with five different samples. The mean value was calculated by the $t$ test (mean ± s.e.m. $n = 6$). *$p < 0.05$, **$p < 0.01$, compared with the NT-PNs group. **b** Cell apoptosis and necrosis analyzed by flow cytometer with Annexin V-FITC/PI double staining after different treatments. **c** Statistical analysis for the percentage of apoptotic cells and necrotic cells in **b**. The mean value was calculated by the $t$ test (mean ± s.e.m. $n = 3$). *$p < 0.05$, **$p < 0.01$, compared with the control group. **d** Confocal images of 4T1 cells upon different treatments and stained with Calcein-AM/PI. Live cells were stained green with calcein AM, and dead/later apoptotic cells were stained red with PI (scale bar, 50 μm)

accumulation and prolonged tumor retention properties, the self-circulation of drug release and ROS burst was expected to last even more than 24 h after administration with DT-PNs, which was favorable for persistent excessive ROS treating for cancers.

**In vivo anticancer activity**. Encouraged by the superior anti-ancer efficacy of DT-PNs in vitro, we carried out in vivo animal experiments to evaluate the chemodynamic therapeutic efficacy by the dual-targeted self-circulation of CPT release and mtROS burst. In vivo anticancer efficacy of DT-PNs was evaluated by intravenous injection into 4T1 tumor-bearing mice. The mice were divided into six groups, including PBS, NT-PNs, TPP-PNs, free CPT, cRGD-PNs, and the resulting DT-PNs. Various samples were administrated via the tail vein at a CPT concentration of 2 mg/kg. The tumor volumes in each group were monitored during the treating process (Fig. 8d, e). The groups of NT-PNs and TPP-PNs did not show obvious inhibition of tumour growth, in addition, free CPT and cRGD-PNs displayed moderate inhibition, whereas the group of DT-PNs exhibited the most significant inhibition. The images of excised tumors from representative mice showed that the tumor size from the DT-PNs treated group was the smallest. Typically, upon 21 day treatment with DT-PNs, the skin of the tumor sites peeled off and new skin formed, indicating almost complete tumor regression. Furthermore, the inhibition ratios were determined for each group, specially, 81% for DT-PNs, 59% for cRGD-PNs, 17% for TPP-

PNs, 12% for NT-PNs, and 43% for free CPT. The survival rate analysis indicated remarkably extended lifetime of the DT-PNs treating group (Fig. 8f) and the change of mice body weight in different treating groups did not display any observable abnormality (Supplementary Fig. 16). H&E analysis confirmed that DT-PNs treated group exhibited much more pyknotic cells with highly condensed nuclei, which were considered to be apoptotic or dead cells (Fig. 8g)[32].

Furthermore, the in vivo toxicology and potential side effects were investigated systematically[67]. The standard haematology markers including the white blood cells (WBC), red blood cells (RBC), haemoglobin (HGB), haematocrit (HCT), mean corpuscular volume (MCV), mean corpuscular haemoglobin (MCH), mean corpuscular haemoglobin concentration (MCHC), platelets (PLT), mean platelet volume (MPV) and thrombocytocrit (PCT) were measured (Fig. 9a). Compared with the PBS group, all the parameters in the five treated groups appeared to be normal and the differences between them were not statistically significant ($p$ value > 0.05). These results indicated that these treatments did not cause obvious infection and inflammation in the treated mice.

Blood biochemical analysis were carried out and various parameters including alanine transaminase (ALT), aspartate transaminase (AST), alkaline phosphatase (ALP), total protein (TP), albumin (ALB), globulin (GLOB), creatinine (CR), blood urea nitrogen (BUN), uric acid (UA) and total bilirubin (TBIL) were examined (Fig. 9b). Compared with the PBS group, no

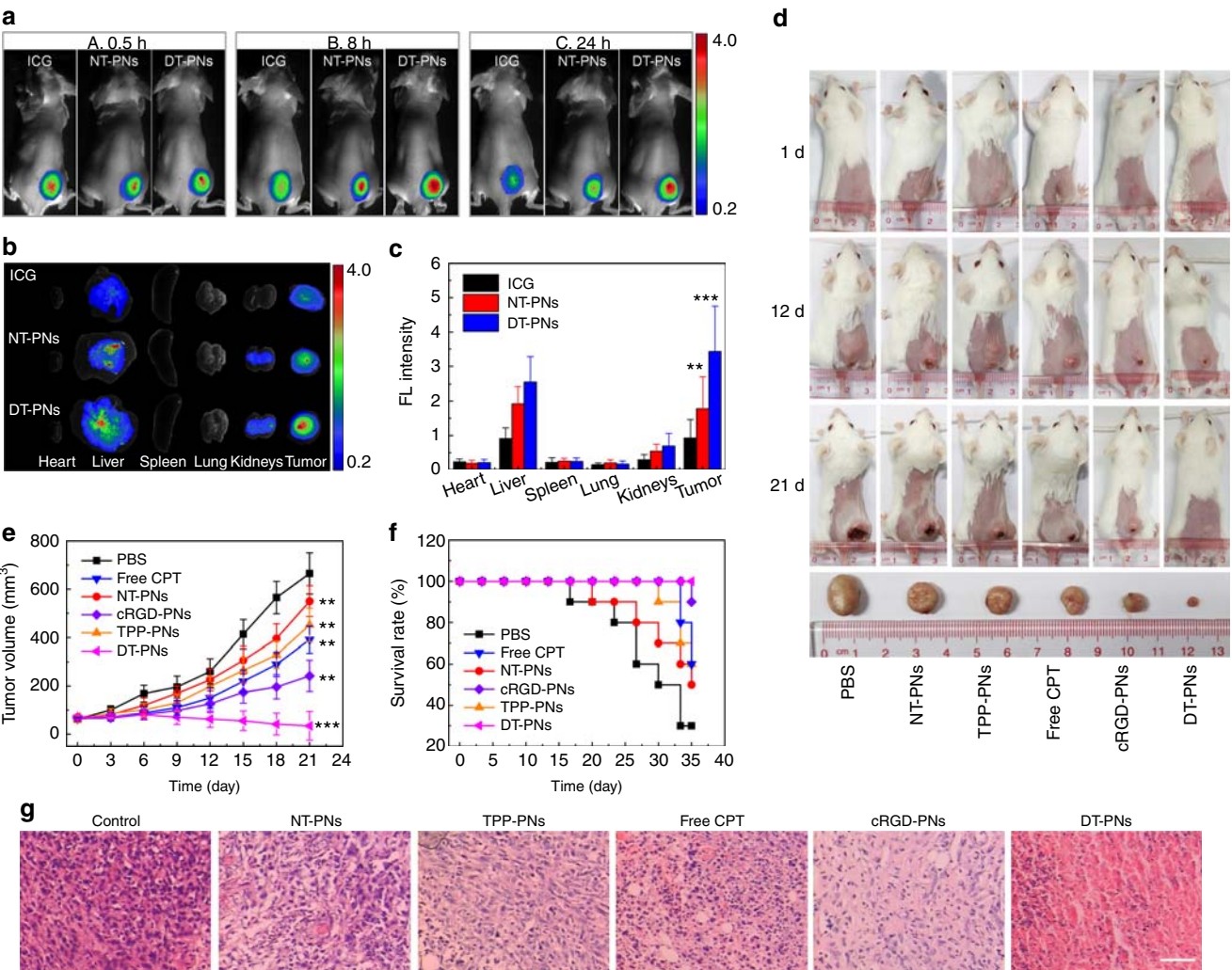

**Fig. 8** In vivo biodistribution and cancer inhibition evaluation. **a** Time-lapse NIR fluorescence images of the tumor after tail vein injection with free ICG, NT-PNs and DT-PNs loaded with ICG, respectively. **b** Fluorescence images of major organs and tumors upon 24 h post-injection with three samples. **c** The biodistribution of free ICG, NT-PNs, and DT-PNs in mice determined by the averaged fluorescence intensity of each organ and tumor. The mean value was calculated by the $t$ test (mean ± s.e.m. $n$ = 5). **$p < 0.01$, ***$p < 0.001$, compared with the ICG group. **d** Representative photographs of mice after different treatments, the bottom tumor obtained from each treating group at 21 d. **e** Tumor volume and **f** survival rate of 4T1 tumor-bearing mice after different treatments. The mean value was calculated by the $t$ test (mean ± s.e.m. $n$ = 6). **$p < 0.01$, ***$p < 0.001$, compared with the PBS group. **g** Representative images of H&E stained tumor sections from different groups (scar bar, 50 μm)

meaningful difference was detected from the five treated groups. Hence, the treatment did not affect the blood chemistry of mice. Furthermore, since alanine transaminase (ALT), aspartate transaminase (AST) and creatinine (CR) are closely related to the functions of the liver and kidney of mice, the results demonstrated that the treatment induced no obvious hepatic and kidney toxicity in mice.

Finally, the corresponding histological changes of organs were checked by immunohistochemistry using major organs including the kidney, lung, spleen, liver and heart collected and sliced for H&E staining (Supplementary Fig. 17). No noticeable signal of organ damage could be observed from all the groups suggesting the test dose were excellently tolerable and had no detectable acute side effects to the examined mice.

## Discussion

Oxidative stress responsive systems are widely investigated in cancer theranostics based on the slightly elevated level of ROS stress in cancer cells (~100 μM)[68], but the therapeutic efficiency is limited most frequently by the relatively insufficient endogenous ROS content in cancer cells and the short lifespan, diffusion distance and action range of ROS[69]. In addition, mitochondria are the major sites for ROS production in aerobic cells, accounting for the consumption of more than 90% cellular oxygen to produce intracellular ROS, primarily superoxide and hydrogen peroxide during respiration. Thus, it is reasonable that mitochondria are recognized as subcellular organelle targets for site-specific drug delivery and therapy. Furthermore, proper level of ROS stress is favorable to promote the fast proliferation and migration of cells, however, excessive oxidative stress can definitely trigger the damage of cancer cells, which is expected to potentiate the therapeutic efficiency in traditional cancer therapy.

Here CPT is selected as a model mitochondria drug to construct ROS-responsive prodrug monomer, CPTSM, then we have developed polyprodrug nanoreactors, DT-PNs, via the facile polymerization of CPTSM to afford polyprodrug amphiphiles and their aqueous self-assembly. DT-PNs demonstrated cancer cell and mitochondria dual-targeting property as well as in situ ROS-responsive CPT release in mitochondria. Endogenous mtROS initiates premier free CPT

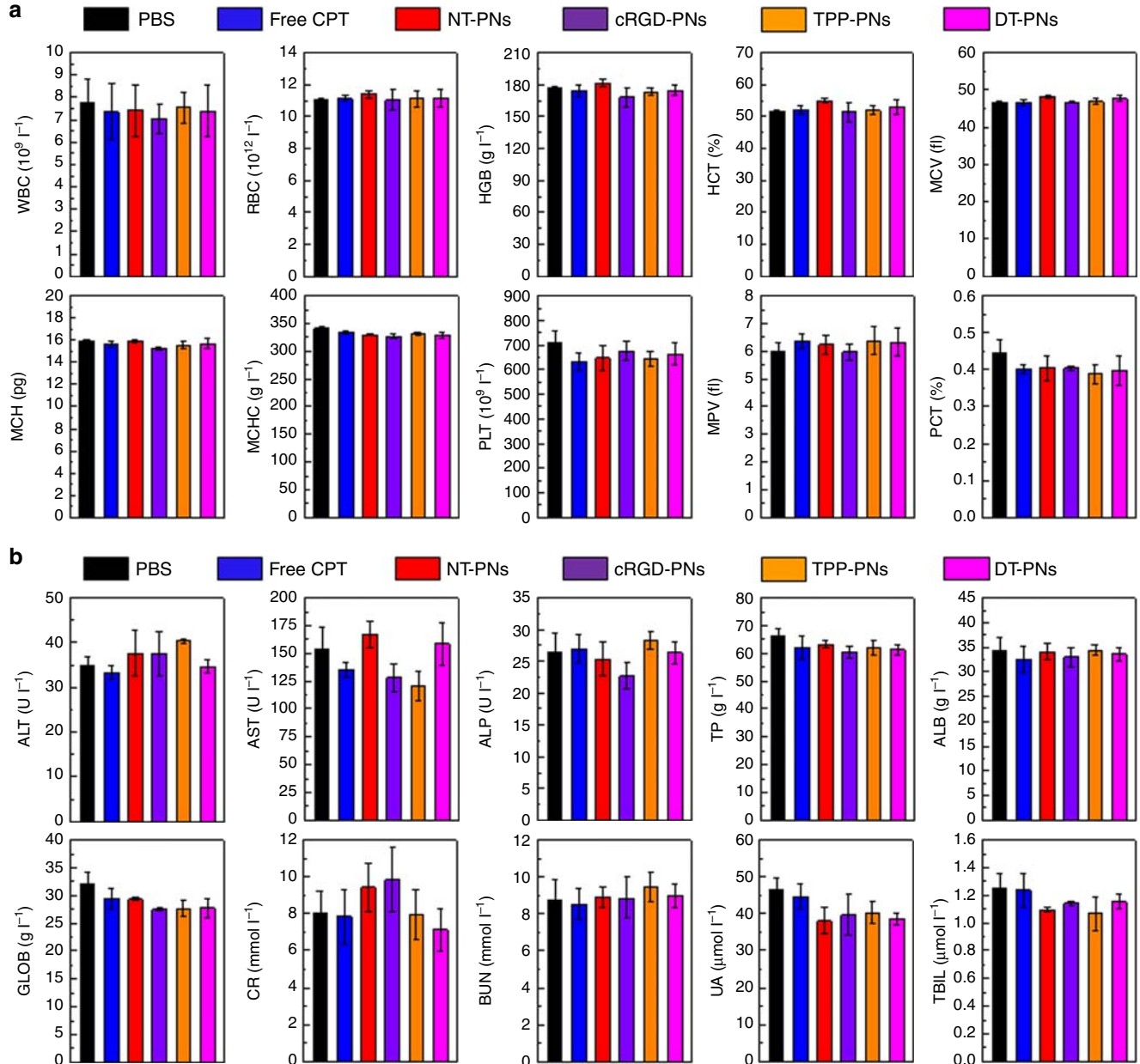

**Fig. 9** In vivo side effects evaluation. **a** Haematological data of the mice intravenously injected with different samples at the 21st day post-injection. The terms are noted as followed: white blood cells (WBC), red blood cells (RBC), haemoglobin (HGB), haematocrit (HCT), mean corpuscular volume (MCV), mean corpuscular haemoglobin (MCH), mean corpuscular haemoglobin concentration (MCHC), platelets (PLT), mean platelet volume (MPV), and thrombocytocrit (PCT). **b** Blood biochemical analysis at the 21st day post-injection. The terms are following: alanine transaminase (ALT), aspartate transaminase (AST), alkaline phosphatase (ALP), total protein (TP), albumin (ALB), globulin (GLOB), creatinine (CR), blood urea nitrogen (BUN), uric acid (UA), and total bilirubin (TBIL)

release at the beginning, followed by activating the cycle of mtROS burst and self-promoted CPT release in mitochondria. Kinetic monitoring suggests that the peak value of mtROS burst happens for cancer cells at ~10 h upon treating with DT-PNs. In the self-circulation, excessive mtROS can be applied to efficiently eliminate cancer cells via the mitochondria-dependent apoptosis (Fig. 10). In summary, three advantageous aspects exist in current design: (1) The polyprodrug and dual targeting strategy to fabricate DT-PNs not only endows the system with ROS-responsiveness, but also the high CPT loading content lays a solid foundation for long-term in situ production of high-dosage ROS in mitochondria[25–27,70]. It is favorable for extended and persistent eradication of cancers based on current chemodynamic modality.[12] (2) The chemodynamic therapeutic activation initiated by endogenous mtROS to afford the resultant

mtROS burst can avoid the limitation of exogenous light in traditional photodynamic therapy, such as penetration depth, limited light exposure area, and some uncertain signal regulation suffered from light irradiation[47]. (3) This design strategy is also promising in the therapeutics of other ROS-related diseases, such as bacterial infections and inflammation, not merely cancer treatments. We anticipate that endogenously activated therapeutic amplification by polyprodrug nanoreactors contributes to the development of theranostics for efficient disease treatment.

## Methods

**In vitro ROS-responsive drug release**. The degradation of ROS-polyprodrug nanoreactors was investigated by incubating DT-PNs (1 mg in 0.5 mL, 0.82 μmol of thioketal groups) with different concentrations of $H_2O_2$, hydroxyl radicals as well

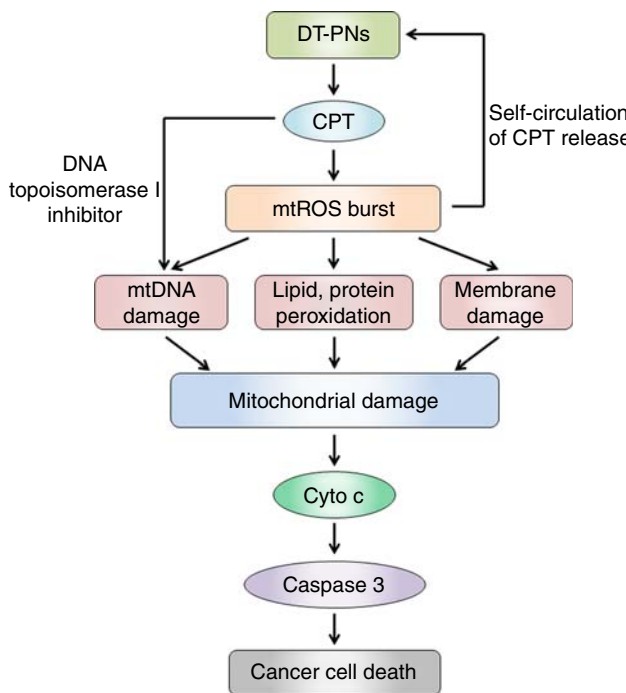

**Fig. 10** Proposed mechanism for the mitochondria-specific self-circulation of CPT release and mtROS burst to damage mitochondria and initiate cell apoptosis/death by dual-targeted polyprodrug nanoreactors (DT-PNs)

as superoxide anion radicals. Superoxide anion radicals were generated by potassium superoxide ($KO_2$) dissolved in degassed PBS (pH 7.0) and hydroxyl radical was formed by mixing Fenton's Reagent with $H_2O_2$. Sodium hypochlorite (NaOCl) at pH 6.02 to form hypochlorous acid ($ClO^-$), $C_{ClO^-} = Abs_{292nm} /0.39$ (mM) and $ONOO^-$ was prepared according to the reported method, $C_{ONOO^-} = Abs_{302nm} /1.67$ (mM).[46] HPLC analysis was used to study the ROS-responsive behavior of the polyprodrug. A 1 mg/mL solution of PDMA-$b$-CPTSM in PBS was prepared and added to an equal volume of PBS containing either: 1 mM ·OH, 1 mM $H_2O_2$, 1 mM $KO_2$, 1 mM $ClO^-$, and 1 mM $ONOO^-$, respectively. After 12 h, 20 μL of the solution was taken for HPLC analysis. In vitro CPT release was performed in culture medium with diverse content of $H_2O_2$ (PBS buffer, pH 7.4) using dialysis membrane tubes (molecular weight cutoff: 3,500 Da). The aqueous dispersion of DT-PNs (1 mg in 0.5 mL, 0.8 μmol of thioketal groups) was placed in the medium and introduced into the dialysis membrane tubes, which were then placed in excessive medium (40 mL) and incubated at 37 °C with different contents of $H_2O_2$ (1, 0.1, 1, and 10 mM). At predetermined intervals, the release medium was collected and then replaced with an equal volume of fresh medium. The CPT content in the release medium was measured by HPLC analysis via the standard curve method at 370 nm of CPT. The particle diameter was obtained from the ImageJ analysis of TEM images, $n = 100$.

**Cell culture**. 4T1 rat breast cancer cells (high integrin $\alpha_v\beta_3$ expression), MCF-7 human breast cancer cells (low integrin $\alpha_v\beta_3$ expression) and U87 human glioblastoma cells were purchased from ATCC and cultured in Dulbecco's modified Eagle's medium (DMEM), respectively. The media was supplemented with 10% fetal bovine serum (FBS) and 1% penicillin−streptomycin in 5% $CO_2$ at 37 °C in a humidified incubator. Before each experiment, the cells were precultured until confluence was reached.

**Cancer cell targeting of DT-PNs**. To study the integrin targeting property of DT-PNs, 4T1 cells and MCF-7 cells were incubated in a serum-free medium containing DT-PNs for 2 h and then rinsed with PBS and replaced with fresh cell medium, respectively. To further validate that the cancer targeting property of DT-PNs based on the decoration of cRGD moieties, preblocking experiments were designed with cells' incubation with 2 μM free cRGD for 30 min before their incubation with DT-PNs. The cells were imaged by a commercial laser scanning microscope (LSM 510/ConfoCor 2) combination system (Zeiss, Jena, Germany), equipped with a Plan-Neofluar 40 × /1.3 NA Oil DIC objective. The channel of RhB-labelled DT-PNs was recorded at the excitation wavelength of 543 nm, and the fluorescence emission was recorded by a LP590 nm filter.

**Cellular uptake analyzed by flow cytometry analysis**. To understand the targeting effect of polyprodrug nanoreactors, 4T1 cells were seeded on 6-well plates at

a density of $1 \times 10^5$, respectively. After incubation for 24 h, the cells were treated with various samples and incubated for 6 h at 37 °C. Then, the cells were washed with cold PBS, harvested and analyzed immediately using flow cytometry (Cytomics FC 500, Beckman Coulter, USA). The mean fluorescence intensity of $1 \times 10^4$ cells was recorded for each sample.

**In vivo ROS fluorescence imaging**. 4T1 tumor-bearing mice were intravenously injected with free CPT, NT-PNs, cRGD-PNs, TPP-PNs and DT-PNs for 8 h, NAC (20 mM) and Vc (10 mM) were intratumor injection before 2 h treatment. Then intraperitoneal administration of ROS probe (Cellular Reactive Oxygen Species Detection Kit, Deep Red Fluorescence)[60] was given, and imaged at 1, 2, and 4 h post-administration.

**Colocalization analysis with mitochondria**. CLSM imaging was performed to observe the co-localization of DT-PNs with mitochondria in different intervals. Briefly, 4T1 cells and U87 cells were seeded in chambered coverslips at a density of $1 \times 10^3$ cells. After incubation for 24 h, the cells were treated with RhB-labeled DT-PNs. At predetermined intervals, cells were washed twice with cold PBS and then incubated with Mitotracker green or lysotracker green to label mitochondria and lysosome according to the manufacturer's protocol, respectively. Then cells were imaged by the CLSM imaging system with identical settings.

**In vitro cytotoxicity assay**. In vitro cytotoxicity was assessed by MTT assays. 4T1 cells were first cultured in DMEM supplemented with 10% fetal bovine serum (FBS), 1% penicillin-streptomycin in 5% $CO_2$ incubator for 24 h. For cytotoxicity assay, 4T1 cells were seeded in a 96-well plate at an initial density of 5000 cells/well in 100 μL of complete DMEM. After incubating for 24 h, DMEM was replaced with fresh medium, and the cells were treated with samples at varying contents (20 mM NAC and 10 mM Vitamin C). The treated cells were incubated in a humidified environment with 5% $CO_2$ at 37 °C for 36 h. After that, MTT reagent (in 20 μL PBS, 5 mg/mL) was added to each well. The cells were further incubated for 4 h at 37 °C. The medium in each well was then removed and replaced by 180 μL DMSO. The plate was gently agitated for 15 min to dissolve the formazan crystals, and the absorbance at 570 nm was recorded by a microplate reader. Each experiment condition was done in quadruple and the data were shown as the mean value plus a standard deviation (±SD).

**Cell apoptosis and necrosis analysis**. 4T1 cells were cultured in 60-mm dishes after different treatments 24 h, then trypsinised, washed with PBS and centrifuged at 3000 rpm for 5 min. Then, cells were resuspended in 500 μL binding buffer and stained with Annexin V-FITC and PI according to the protocol. The cells were incubated in the dark at room temperature for 15 min. Finally, the percentage of apoptotic cells and necrotic cells were assessed by FACScan flow cytometry.

**Mitochondrial membrane potentials assay**. JC-1 probe was employed to evaluate the mitochondrial depolarization in 4T1 cells. Briefly, cells cultured in a 35 mm confocal dish after indicated treatments 12 h were incubated with an equal volume of serum-free medium containing JC-1 dye (5 mg/L) at 37 °C for 20 min and rinsed twice with PBS, then placed in fresh medium without serum. Finally, images were taken in the green and red fluorescence channel by CLSM imaging. The images were obtained at 488 nm excitation and 530 nm emission for green (JC-1 monomers) and at 543 nm excitation and 590 nm emission for red fluorescence (JC-1 aggregates).

**Measurement of cellular ATP levels**. 4T1 cells were respectively seeded at a density of $2 \times 10^5$ cells per well and then incubated for 24 h prior to experiments. Cells were then treated with DT-PNs (2 mg/mL) for a particular time (12 and 24 h) in DMEM media supplemented with 10% FBS and 1% antibiotics at 37 °C. Thereafter, the cells were washed with PBS, harvested with trypsin for 1 min at 37 °C. Cold PBS was added to terminate the reaction and cells were collected by centrifugation. ATP level was assessed using the adenosine 5'-triphosphate (ATP) bioluminescent assay kit (Sigma-Aldrich) and a chemiluminometer (Lumat LB9507, EG&G Berthold)

**Western blot analysis**. 4T1 breast cells were seeded in 6-cm petri dishes at a density of $1 \times 10^6$ and cultured for 24 h. Then, DT-PNs (5 μg/mL) were added and the cells were allowed to culture for 12 h and 24 h, respectively. The cells were washed with precooled PBS for three times before being scraped. Proteins were extracted from the cells using 100 μL lysis buffer (50 mM Tris–HCl pH 8.0, 150 mM NaCl, 1% Triton X-100, 100 mM PMSF). The samples were separated by 15% SDS-PAGE gel electrophoresis and transferred to PVDF membranes (Millipore, Bedford, MA, USA). The membrane was blocked in TBST (10 mM Tris-HCl, pH 7.4, 150 mM NaCl and 0.1% Tween-20) containing 5% skim milk, incubated with the indicated antibodies and the secondary antibodies. The signals were detected with an LI-COR Odyssey Scanning Infrared Fluorescence Imaging System (LI-COR, Lincoln, NE, USA). The following antibodies were used for immunoblot: rabbit monoclonal anti-cytochrome c antibody (1:1000 dilution, Cell Signaling Technology), rabbit monoclonal anti-caspase-3 antibody (1:1000 dilution, Cell

Signaling Technology), mouse monoclonal anti-Bcl-2 antibody (1:1000 dilution, Cell Signaling Technology), mouse monoclonal anti-Bax antibody (1:1000 dilution, Cell Signaling Technology) and moue monoclonal GAPDH antibody (1:1000 dilution, Cell Signaling Technology), IRDye 800 CW anti-rabbit IgG (1:3000 dilution, Santa Cruz Technoogy), Alexa Fluor 680 goat anti-Mouse IgG (1:3000 dilution, Santa Cruz Technoogy). The pretreatment with NAC or Vc was employed to eliminate cellular ROS as two control groups.

**Mitochondrial fractionation**. After different treantments, 4T1 cells were harvested and then fractionated according to the guideline for the cell mitochondria isolation kit (Beyotime Institute of Biotechnology, Haimen, China). Briefly, cells were resuspened in a mitochondria extraction reagent (provided in the kit) and homogenized with a microhomogenizer, then placed in ice bath for 15 min. The homogenates were centrifuged at 600 g for 10 min at 4 ℃. The supernatants were collected and further centrifuged at 11,000 g for 10 min at 4 ℃. The cytosol fraction was collected from the supernatant and the western blotting was used to validate the translocation of Cyto c from mitochondria to cytosolic milieu. Finally, The mitochondrial fraction was collected from the precipitates, and the ROS content in mitochondria was used to quantified by flow cytometer analysis.

**Animal xenograft model**. Female Balb/c mice (5-weeks-old) were purchased from Center for Experimental Animals, Southern Medical University. All animal studies were conducted in accordance with the guidelines of the National Regulation of China for Care and Use of Laboratory Animals (South China Normal University, Guangzhou, China). 4T1 tumor-bearing mouse model was successfully established by subcutaneous injection of $4 \times 10^6$ cells suspended in 100 µL PBS into the right axilla of each mouse. The mice were treated when the tumor volumes approached 60~70 mm³. The 4T1 tumor-bearing mice were randomly distributed into six groups, and 100 µL of the corresponding CPT was caudal vein injected into the tumor-bearing mice and injected every 2 days: (1) control group (only treating with PBS); (2) NT-PNs; (3) TPP-PNs; (4) cRGD-PNs; 5) Free CPT; 6) DT-PNs, respectively. All groups received only one sample injection. The therapeutic results of each group were evaluated by measuring the tumor volumes after 21 days. Tumor volume (V) = length × width$^2$/2. Relative tumor volume was calculated as V/V$_0$ (V$_0$ was the corresponding tumor volume when the treatment was initiated).

**In vivo biodistribution analysis**. The 4T1 tumor-bearing mice were removed hairs from skins and randomly divided into three groups ($n = 3$, each group). The mice were intravenously injected with 100 µL PBS, free ICG and ICG-loaded DT-PNs (both containing 80 µg/mL ICG). Fluorescent images were taken at 0.5, 8, and 24 h post-injection using a 808 nm excitation laser and a 810 nm filter to collect the fluorescence signals. The mice were sacrificed at 24 h post-injection. Then the organs including heart, liver, spleen, lung, kidneys, and tumor were collected for imaging and biodistribution analysis.

**Statistical analysis**. All data were representative results from at least three independent experiments and mean ± s.e.m. were shown. Statistical analyses were performed using the t test. *$p < 0.05$, **$p < 0.01$ and ***$p < 0.001$ were considered statistically significant.

**Reporting summary**. Further information on experimental design is available in the Nature Research Reporting Summary linked to this article.

## Data availability
The data in this work are available in the manuscript or Supplementary Information, or available from the corresponding author upon request.

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

## Acknowledgements

This work was supported by the National Natural Scientific Foundation of China (NSFC, 21674040, 81630046), the Natural Science Foundation for Distinguished Young Scholars of Guangdong Province (2016A030306013), the Guangdong Program for Support of Top-notch Young Professionals (2015TQ01R604), the Science and Technology Planning Project of Guangdong Province (2015B020233016, 2014B020215003), the Scientific Research Projects of Guangzhou (201805010002), and the National Key Research and Development Program of China (2018YFA0209800).

## Author contributions

X.L.H. conceived the project. X.L.H. and D.X. supervised and supported the project. X.L.H. and W.J.Z. designed and conducted the experiments and wrote the manuscript. Q.S. performed the western blot analysis. All authors discussed the results and commented on the manuscript.

## Additional information

**Competing interests:** The authors declare no competing interests.

