## [Peer Review File · Nature Communications]

Reviewers' comments:

Reviewer #1 (Remarks to the Author):

The current work attempts to report that nanoparticles of thioketal-bridged prodrug polymers could release CPT drug under initial ROS action within mitochondria, then the release of CPT will render even higher ROS levels by using cellular mechanisms, then cycled ROS generation will eventually release all CPT drugs from nanoparticles. My main concerns are described below:

1) ROS within cells exists in various forms such as hydrogen peroxide, singlet oxygen, superoxide radical, peroxyxynitrite, etc. All of them are at very low levels (i.e., constantly generating and consuming). In the introduction part, the authors mainly discuss ROS, but they eventually work with hydrogen peroxide. Throughout the main text, the establishment of cycled ROS generation or H₂O₂ generation upon cellular internalization of CPT NPs has not been achieved, no direct evidences were given. Note that even cancer cells can not constantly generate hydrogen peroxide levels higher than 1 mM.

2) Previously, quite a few reports deal with thioketal linkages conjugated with either drugs or nanoparticle degradation. For example, in the work of Farokhzad et al. 10.1002/adma.201700141, KO₂ (much stronger oxidation potential than hydrogen peroxide) was used to release MTO drug. In the work by Xia et al (10.1002/anie.201209633), 50-200 mM hydrogen peroxide was used to trigger the degradation of thioketal-containing NPs. In the work by Liu et al. (Angew. Chem. Int. Ed. 53, 7163), in situ generated singlet oxygen was used to cleave thioketal linkage. In the original work by Murthy et al. (10.1038/NMAT2859), again, KO₂ was used to cleave thioketal linkage. Although the authors have cited these references, they failed to compare the reaction conditions for thioketal. In the current work, the authors reported that ~1 mM hydrogen peroxide could cleave most of thioketal linkage (Figure 2d), which contradict with previous reports. The release of CPT also lacks direct support of HPLC or LC-MS data from both the polymer and model compounds.

3) Previous organic chemistry literature reports (Synthetic Communications1, 41: 2374-2384; <http://www.organic-chemistry.org/protectivegroups/carbonyl/1,3-dithiolanes.htm>) 2011) tell us that these thioketal linkages are very stable towards low level of hydrogen peroxide ("neither 30%H₂O₂ nor 20 mol%NH₄I alone was effective as a cleaving reagent, Table 1).

4) Another major concern is that the stated "which can be activated by initial endogenous mtROS to release slight free CPT in mitochondria. The in-situ released CPT instantly acts as a cellular respiration inhibitor to induce." in the abstract lacks direct experimental evidences at all. This is quite speculative and not good for scientific research practices.

Reviewer #2 (Remarks to the Author):

This manuscript by Zhang and colleagues describes the development of a new therapeutic approach to treat cancer. Namely they describe the development and application of mitochondria targeting nano-reactors, targeting camptothecin (CPT) to the mitochondria. The authors use integrin (RGD) targeting motifs to enable cancer cell specific targeting of these pro-drugs – they propose that mitochondrial ROS frees camptothecin within the mitochondria whereupon it can activate more ROS (acting in a feed forward manner), ultimately the ROS generated is proposed to kill via mitochondrial apoptosis. In doing so, this offers a new way to develop and enhance cancer therapy. While I found this study novel and exciting, in my view, the authors often over interpret their data and a lot is drawn from correlation, where definitive expts. should be applied. Specific aspects I think should be addressed are the following.

- while addition of hydrogen peroxide in vitro catalyses the release of CPT from the pro-drug, it is unclear whether this occurs in cells. I think this needs to be addressed, the in vitro conditions whereby release is seen uses high concs. of H₂O₂ and prolonged incubation, as such its unclear to me whether ROS driven release of CPT occurs in cells, and by implication is required for these nanoreactors to exert their cytotoxic effects.

- the authors imply that the accumulation of DT-NPs in cells in cancer specific due to α1B3 integrin expression (Fig 3). Its important 1) to examine whether DT-NPs also accumulate in non-transformed cells in vitro (i.e. is it cancer specific) 2) test whether α1B3 is relevant for the

accumulation they observe, e.g. does RNAi/CRISPR deletion of $\alpha 1B3$ in 4T1 cells block DT-NP entry into cells.

- the authors correlate ROS with mitochondrial apoptosis/cell death (Fig 5). The data showing increase in mitochondrial ROS after DT-NP especially is very impressive, however at no point is the production of ROS (as a means to kill cells) directly tested i.e. do ROS scavengers block DT-NP cytotoxicity. If DT-NP is killing via disruption of mitochondrial function, then ROS would be a consequence rather than a cause of cytotoxicity – this needs to be directly examined.

- the implication that ROS activates mitochondrial apoptosis is rather weak (Fig 5d – note its unclear what the treatments are in this figure). This needs to be better investigated, showing not only cytoplasmic cyt C expression but the corresponding mitochondrial fractions. More importantly, does inhibition of mitochondrial apoptosis (e.g. by overexpressing Bcl2) block DT-NP induced killing.

- the in vivo expts. investigating efficacy are impressive (Fig 7), not only in the potent tumour killing (by DT-NP) but the lack of apparent cytotoxicity. With respect to the latter, given the significant accumulation in both liver and kidney I think it is important to investigate parameters of toxicity in these organs.

- Fig 8/discussion thereof – its important to note that mitochondrial depolarisation does not equate with mitochondrial permeabilisation (leading to cyt c release) the proposed model should be modified accordingly.

Other points

- while readable, the ms. could do with extensive editing to improve its flow/grammar.

Reviewers' comments:

Reviewer #1:

The current work attempts to report that nanoparticles of thioketal-bridged prodrug polymers could release CPT drug under initial ROS action within mitochondria, then the release of CPT will render even higher ROS levels by using cellular mechanisms, then cycled ROS generation will eventually release all CPT drugs from nanoparticles. My main concerns are described below:

(1) ROS within cells exists in various forms such as hydrogen peroxide, singlet oxygen, superoxide radical, peroxyxynitrite, etc. All of them are at very low levels (i.e., constantly generating and consuming). In the introduction part, the authors mainly discuss ROS, but they eventually work with hydrogen peroxide. Throughout the main text, the establishment of cycled ROS generation or H₂O₂ generation upon cellular internalization of CPT NPs has not been achieved, no direct evidences were given. Note that even cancer cells can not constantly generate hydrogen peroxide levels higher than 1 mM.

Response: Thanks for the reviewer's constructive comments. The reviewer is right! Endogenous ROS content of cancer cells is not very high; whereas in this work, *the slightly high expression of inherent ROS was the trigger to initiate slight CPT release in mitochondria at the beginning*, where CPT acted as a cellular respiration inhibitor to *stimulate the regeneration of much more mtROS*. Finally, much more CPT release and significantly enhanced oxidation stress would damage cancer cells efficiently (Fig. 1). Herein, Current chemodynamic strategy of endogenously activated ROS amplification in mitochondria *not only overcomes the short lifespan and action range of low-level endogenous ROS, but also avoids the limitation*

of exogenous light in typical photodynamic therapy.

As suggested by the reviewer, three kinds of ROS species, hydroxyl radical ($\cdot\text{OH}$), superoxide radical, and H_2O_2 were employed to examine the ROS-responsive drug release property from thioketal-bridged polyprodrug, respectively. The results demonstrated typical CPT release based on the HPLC analysis (Figure 2c). *In vitro quantitative CPT release was determined for these three kinds of ROS species*, in which the efficiency of hydroxyl radical was highest. In addition, singlet oxygen could also cleave thioketal linkages, which was also demonstrated by traditional photodynamic process (*Angew. Chem. Int. Ed.* 2014, 53, 7163-7168; *Biomaterials*, 2018, 171, 72-82; *Theranostics*, 2018, 8, 2939-2953).

The in-situ released CPT in mitochondria could act as cellular respiration inhibitor to induce intracellular ROS burst, please kindly refer to the updated discussion at page 7 and page 8: “Subsequently, the ROS state of 4T1 cells was evaluated upon treating with four kinds of polyprodrug nanoparticles with different targeting moieties and free CPT, respectively. *The overall intracellular ROS was firstly evaluated by flow cytometry analysis (FACS) using 2',7'-dichlorofluorescein diacetate (DCFH-DA), which could be rapidly oxidized by ROS to generate green fluorescent dichlorofluorescein (DCF). Notably, sharply increased DCF fluorescence was detected in the DT-NPs group after 8 h incubation. In contrast, the fluorescence intensity change of NT-NPs, cRGD-NPs and TPP-NPs groups was not obvious (Fig. 5a, b). In addition, the intracellular total ROS up-regulation induced by DT-NPs could be significantly attenuated by two ROS scavengers (20 mM N-acetylcysteine, NAC and 10 mM Vitamin C, Vc)^{48,49}, which indicated that much more CPT accumulation in mitochondria based on the dual-targeting property could significantly enhance the total intracellular ROS level. Such ROS burst of DT-NPs was further visualized with confocal laser scanning microscopy (CLSM) imaging (Fig. 5c), which was in good agreement with the results shown in Fig. 5a, b.*

Furthermore, the specific mitochondrial superoxide, the source of most ROS species, was also evaluated upon different treatments. FACS and CLSM imaging was employed to discern the superoxide radical in mitochondria using MitoSOX Red as an indicator (Fig. 5d, e, f)⁵⁰. Upon treating with DT-NPs for 8 h, highest red fluorescence was observed, demonstrating highest mitochondrial superoxide generation, potentially endowed by the in-situ mitochondrial CPT release. Herein, the highest level of superoxide in mitochondria would guarantee the formation of much more amount of ROS species in mitochondria. Thus for cells treated with DT-NPs, the lipophilic and positively charged TPP moiety could efficiently promote the accumulation of DT-NPs in negatively charged mitochondria, in which remarkable mtROS was subsequently activated due to the in-situ self-promoted CPT release from DT-NPs in mitochondria.

Finally, we further interrogated that whether such ROS burst induced by DT-NPs could be realized in vivo. For this aim, 4T1 tumor-bearing mice were intravenously injected with different formulations for 8 h and then intraperitoneal administration of ROS probe, Cellular Reactive Oxygen Species Detection Assay

Kit (Deep Red Fluorescence)⁵¹, and imaged at 1 h, 2 h and 4 h post-administration (Fig. 5g). Interestingly, the fluorescence intensity of tumor sites (white arrow indicated) of free CPT, NT-NPs, cRGD-NPs and TPP-NPs groups exhibited unobvious enhancement at 1 h, 2 h and 4 h. For the evaluation of intracellular total ROS level and specific mitochondrial superoxide, the free CPT group showed moderate enhancement (Fig. 5a-f), but for *in vivo* ROS evaluation, the extent of ROS increasing in tumor sites was comparatively unobvious, potentially due to its poor water solubility, instability and compromised pharmacokinetics⁵². In contrast, for the group treated with DT-NPs, ~12.6-fold fluorescence intensity was detected at 4 h compared with the control group, suggesting significant ROS level in tumor sites (Fig. 5f). Furthermore, the tumor fluorescence intensity for the DT-NPs group could be significantly attenuated in the presence of ROS scavengers, NAC and Vc. Thus, we envisaged that efficient tumor accumulation of DT-NPs and subsequent mitochondria-specific ROS burst contributed to the *in vivo* tumor ROS increasing.”

Figure 2c. HPLC traces recorded for PDMA-*b*-CPTSM upon treating with H₂O₂, KO₂ and hydroxide radical (\cdot OH). The mobile phase was 50/50 acetonitrile and water at a flow rate of 1.0 mL/min.

Figure 2d. *In vitro* CPT release from DT-NPs (1 mg, 0.8 μ mol equivalent thioketal group) in the presence of H₂O₂, potassium superoxide (KO₂) or hydroxide radical.

Figure 5 | *In vitro* and *in vivo* demonstration of ROS burst triggered by DT-NPs. (a) Determination of intracellular total ROS level for 4T1 cells upon different treatments for 8 h detected by DCFH-DA based on flow cytometry analysis. (b) Statistical analysis of the mean fluorescence intensity in (a). (c) CLSM imaging of intracellular total ROS by DCFH-DA staining upon incubation with DMEM medium (control), free CPT, NT-NPs, cRGD-NPs, TPP-NPs, and DT-NPs with/without 20 mM NAC or 10 mM Vc for 8 h, respectively. (d) Determination of superoxide in mitochondria upon different treatments for 8 h detected by MitoSOX based on flow cytometry analysis. (e) Statistical analysis of the mean fluorescence intensity in (d). (f) CLSM imaging of superoxide in mitochondria by MitoSOX staining upon the same treatments in (c). (g) *In vivo* fluorescence imaging of 4T1 tumor-bearing Balb/c mice after intravenous injection with free CPT, NT-NPs, cRGD-NPs, TPP-NPs, and DT-NPs with/without 20 mM NAC or 10 mM Vc for 12 h, and followed by intraperitoneal injection with one ROS probe (Cellular Reactive Oxygen Species Detection Assay Kit, Deep Red Fluorescence), white arrows indicate the tumor sites. (h) Quantitative analysis for the fluorescence intensity of tumor sites in (g).

(2) Previously, quite a few reports deal with thioketal linkages conjugated with either drugs or nanoparticle degradation. For example, in the work of Farokhzad et al. 10.1002/adma.201700141, KO_2 (much stronger oxidation potential than hydrogen peroxide) was used to release MTO drug. In the work by Xia et al (10.1002/anie.201209633), 50-200 mM hydrogen peroxide was used to trigger the degradation of thioketal-containing NPs. In the work by Liu et al. (Angew. Chem. Int. Ed. 53, 7163), in situ generated

singlet oxygen was used to cleave thioketal linkage. In the original work by Murthy et al. (10.1038/NMAT2859), again, KO_2 was used to cleave thioketal linkage. Although the authors have cited these references, they failed to compare the reaction conditions for thioketal. In the current work, the authors reported that ~ 1 mM hydrogen peroxide could cleave most of thioketal linkage (Figure 2d), which contradict with previous reports. The release of CPT also lacks direct support of HPLC or LC-MS data from both the polymer and model compounds.

Response: Thanks for the reviewer's detailed comments and helpful suggestions. Firstly, KO_2 , hydroxyl radical and hydrogen peroxide were performed as the trigger to mediate CPT release, which was determined by HPLC analysis (Figure 2c, 2d). Please kindly refer to **Response 1**.

Secondly, we are sorry to miss some experimental detail. In fact, the molar content of thioketal linkages is $0.82 \mu\text{mol}$ (2 mg/mL , 0.5 mL) in the *in vitro* drug release experiment. Whereas the content of thioketal group is $10.82 \mu\text{mol}$ in the work by Xia et al. (10.1002/anie.201209633, *Supporting Information, Page 3*). The relative content of thioketal groups is ~ 13.5 -fold in the Xia's work than our manuscript, thus relatively low content of hydrogen peroxide is enough to mediate CPT release in this work, and current results are comparable with previous reports. In addition, hydrogen peroxide was employed to trigger the degradation of thioketal-containing NPs in the work by Liu et al (*Angew. Chem. Int. Ed.* **2014**, *53*, 7163), unfortunately, we didn't find the precise content of thioketal group in the experiment. Just like the work of Prof. Liu's group, hydrogen peroxide could cleave thioketal groups, which was also confirmed by HPLC analysis in this work (Figure 2c).

The main text in the discussion part was updated accordingly in page 4. *"To investigate the ROS-induced CPT release, KO_2 was employed to generate superoxide, and the Fenton reaction between Fe^{2+} and H_2O_2 was used to generate hydroxyl radical ($\cdot\text{OH}$), which were well demonstrated before^{37,38}. HPLC analysis showed that PDMA-*b*-PCPTSM could readily release CPT in the presence of 1 mM KO_2 , 1 mM H_2O_2 and 1 mM $\cdot\text{OH}$, respectively (Fig. 2c). The *in vitro* CPT release rate was further evaluated upon treating with three types of ROS species, including hydroxyl radical, H_2O_2 , and superoxide (Fig. 2d). Hydroxyl radical was observed to mediate the fastest CPT release among three kinds of ROS species, which also agreed with their relative oxidation potency⁴".*

(3) Previous organic chemistry literature reports (*Synthetic Communications*1, 41: 2374-2384; <http://www.organic-chemistry.org/protectivegroups/carbonyl/1,3-dithiolanes.htm>) 2011) tell us that these thioketal linkages are very stable towards low level of hydrogen peroxide ("neither 30% H_2O_2 nor 20 mol% NH_4I alone was effective as a cleaving reagent, Table 1).

Response: Thanks for the reviewer's insightful review. The reviewer is right, as reported in the literature (*Synthetic Communications*, 2011, 41, 2374, Figure R1a), the compound with a six-membered ring thioketal group is stable towards 30% H_2O_2 or 20% NH_4I alone. In organic chemistry, chemical groups

with five-membered ring or six-membered ring are generally much stable. In some other excellent papers, the thioketal groups without five-membered or six-membered ring all exhibits ROS-responsive cleavage property (Figure R1b-e). Compared with the difference among these reported thioketal groups, we envisaged that the six-membered ring structure afforded extra stability and steric hindrance against degradation for the molecule in Figure R1a. Furthermore, the thioketal group with dimethyl groups in this work is similar to these published structures in Figure R1b-e, possessing comparable ROS-cleavable potency (Figure R1f vs Figure R1b-e).

Figure R1. Typical chemical structures with thioketal linkages in reported literatures and this work (a) A compound with a six-membered ring thioketal group, 2-(3,4-methylenedioxyphenyl)-1,3-dithiane, is stable towards 30% H₂O₂ or 20 mol%NH₄I alone. (b)-(e) Reported thioketal groups with ROS-sensitive cleavage property. (f) ROS-responsive polyprodrug tethered with thioketal linkages in current work.

(4) Another major concern is that the stated “which can be activated by initial endogenous mtROS to release slight free CPT in mitochondria. The *in-situ* released CPT instantly acts as a cellular respiration inhibitor to induce...” in the abstract lacks direct experimental evidences at all. This is quite speculative and not good for scientific research practices.

Response: Thanks for the reviewer’s excellent advice. As suggested, further CLSM imaging and line-scan profile analysis were performed to show the ROS-induced CPT release in mitochondria. CLSM imaging of 4T1 cells demonstrated the significant co-localization of DT-NPs (red RhB-labeled polymer backbone), CPT (blue) and mitochondria (green), and significant CPT blue fluorescence was found in mitochondria even after 16 h incubation. The mitochondria targeting property of DT-NPs promoted the fast mitochondrial localization. Due to the small size of mitochondria and the resolution limit of CLSM imaging, the CPT cleavage from the polymer backbone in mitochondria was hard to be observed in

molecular level, thus exhibiting excellent co-localization in mitochondria. ROS-induced CPT release was demonstrated in Figure 2c-f, thus intracellular inherent mtROS could also mediate *in-situ* CPT release in mitochondria. *In vitro* and *in vivo* determination of intracellular total ROS and mitochondrial ROS definitely proved the mtROS burst upon treated with DT-NPs (Figure 5, Please refer to Response 1).

Furthermore, the *in-situ* CPT release in mitochondria could damage mitochondria obviously, exhibiting significant mitochondria fragmentation, and the post-treatment with different antioxidants (NAC and Vc) would prevent the mitochondrial fragmentation to some extent due to the effect of ROS down-regulation (Figure 6b). Finally, the cellular respiration inhibition was confirmed by evaluating the cellular ATP level (Figure 6c). Upon incubation with DT-NPs for 24 h, ~4.2-fold ATP decrease was determined compared with the control group, and the treating with antioxidants could prevent ATP decrease efficiently.

Figure 4 | Mitochondria-specific localization and *in-situ* drug release in mitochondria. 4T1 cells were treated with RhB-labeled DT-NPs for 2 h, 4 h, 8 h, and 16 h, respectively. RhB (red channel) was utilized to label the polymer backbone. Mitotracker (green channel) was employed to co-stain mitochondria. Blue channel was originated from the emission of CPT itself. In the line-scan profiles, the red, blue and green curves represent the fluorescence intensity from DT-NPs, CPT and Mitotracker Green, respectively (scale bar, 5 μm).

Figure 6b. CLSM images for the mitochondrial morphologies in 4T1 cells after 24 h incubation with DT-NPs, the cells were stained with MitoTracker Red (scale bar, 5 μ m).

Figure 6c. 4T1 cells were treated with DT-NPs for 12 h and 24 h for the cellular ATP measurement.

Reviewer #2:

This manuscript by Zhang and colleagues describes the development of a new therapeutic approach to treat cancer. Namely, they describe the development and application of mitochondria targeting nano-reactors, targeting camptothecin (CPT) to the mitochondria. The authors use integrin (RGD) targeting motifs to enable cancer cell specific targeting of these pro-drugs - they propose that mitochondrial ROS frees camptothecin within the mitochondria whereupon it can activate more ROS (acting in a feed forward manner), ultimately the ROS generated is proposed to kill via mitochondrial apoptosis. In doing so, this offers a new way to develop and enhance cancer therapy. While I found this study novel and exciting, in my view, the authors often over interpret their data and a lot is drawn from correlation, where definitive

experiments should be applied. Specific aspects I think should be addressed are the following.

(1) While addition of hydrogen peroxide *in vitro* catalyses the release of CPT from the pro-drug, it is unclear whether this occurs in cells. I think this needs to be addressed, the *in vitro* conditions whereby release is seen uses high concentrations of H₂O₂ and prolonged incubation, as such it's unclear to me whether ROS driven release of CPT occurs in cells, and by implication is required for these nanoreactors to exert their cytotoxic effects.

Response: Thank for the reviewer's insightful comments to improve this work. For the issue of *in-situ* Mitochondrial CPT release and mtROS burst, please kindly refer to **Response 1 and 4** for the first Reviewer. Furthermore, MTT assay and flow cytometer analysis indicated the excellent cytotoxicity of DT-NPs towards 4T1 cells (Figure 7). However, the treatment of ROS scavengers (NAC or Vc) would greatly decrease the cytotoxicity of DT-NPs (Supplementary Figure 11). Herein, mtROS-mediated CPT release in mitochondria and subsequent ROS burst contributed their cytotoxic effects.

Supplementary Figure 11. *In vitro* cytotoxicity determined by MTT assay against 4T1 cells upon 36 h treatment with DT-NPs with or without NAC or Vc.

(2) The authors imply that the accumulation of DT-NPs in cells is cancer specific due to $\alpha_v\beta_3$ integrin expression (Fig 3). It's important 1) to examine whether DT-NPs also accumulate in non-transformed cells *in vitro* (i.e. is it cancer specific) 2) test whether $\alpha_v\beta_3$ is relevant for the accumulation they observe, e.g. does RNAi/CRISPR deletion of $\alpha_v\beta_3$ in 4T1 cells block DT-NP entry into cells.

Response: Thanks for the constructive advice: (1) Integrin $\alpha_v\beta_3$ is not cancer specific, but it is overexpressed in many malignant tumors, which has been well-demonstrated as the target to promote efficient cellular uptake into cancer cells internalization. (2) As suggested by the reviewer, we performed pre-blocking experiments based on CLSM imaging and statistical flow cytometry analysis (Figure 3a, b). Cells were pretreated with excess free cRGD before incubation with DT-NPs, suggesting almost complete inhibition of cellular uptake, which confirmed that $\alpha_v\beta_3$ was relevant with the nanoparticle accumulation.

Figure 3 | Cancer cell targeting property of DT-NPs. (a) CLSM images of 4T1 cells (integrin positive) and MCF-7 cells (integrin negative) after incubating with RhB-labelled DT-NPs under different treatments for 2 h at 37 °C. For the pre-blocking experiment, cells were pretreated with 2 μ M free cRGD for 30 min before incubation with DT-NPs. (b) and (c) Flow cytometry analysis and the statistical analysis in (a). (d) Flow cytometry analysis of 4T1 cells upon 6 h treatment with RhB-labelled polyprodrug nanoparticles with diverse targeting moieties, including non-targeting nanoparticles (NT-NPs) from the assembly of PDMA-*b*-PCPTSM and PDMA-*b*-P(CPTSM-*co*-RhB); TPP-NPs from the assembly of TPP-PDMA-*b*-PCPTSM and PDMA-*b*-P(CPTSM-*co*-RhB), cRGD-NPs from the assembly of cRGD-PDMA-*b*-P(CPTSM-*co*-RhB) and PDMA-*b*-PCPTSM, and the final resultant DT-NPs (scale bar, 50 μ m). (e) Statistical analysis of the mean fluorescence intensity in (b).

(3) The authors correlate ROS with mitochondrial apoptosis/cell death (Fig 5). The data showing increase in mitochondrial ROS after DT-NP especially is very impressive, however at no point is the production of ROS (as a means to kill cells) directly tested i.e. do ROS scavengers block DT-NP cytotoxicity. If DT-NP is killing via disruption of mitochondrial function, then ROS would be a consequence rather than a cause of cytotoxicity – this needs to be directly examined.

Response: We agree with the reviewer’s opinion, i.e., ROS scavengers definitely block the cytotoxicity of DT-NPs. The increase of mitochondrial ROS was demonstrated based on *in vitro* and *in vivo* explorations (Figure 5). Please kindly refer to the **Response 1** for the first Reviewer. MTT assays confirmed that antioxidants (NAC and vitamin C) could eliminate ROS to reduce the cytotoxicity of DT-NPs (Supplementary Figure 11). Furthermore, the *in-situ* CPT release in mitochondria and subsequent ROS burst could damage mitochondria obviously, exhibiting significant mitochondria fragmentation and loss of membrane potential, and the post-treatment with antioxidants (NAC and Vc) would prevent the

mitochondrial fragmentation due to the effect of ROS elimination (Figure 6a, 6b). Please also kindly refer to **Response 1**.

Figure 6a. (a) CLSM images of JC-1 stained cells after different treatments, the fluorescence transition from red to green indicated significant mitochondrial damage (scale bar: 20 μm).

Figure 6b. CLSM imaging of mitochondrial morphology in 4T1 cells after incubation for 24 h with DT-NPs, the cells were stained with MitoTracker Red (scale bar, 5 μm).

(4) The implication that ROS activates mitochondrial apoptosis is rather weak (Fig 5d – note its unclear what the treatments are in this figure). This needs to be better investigated, showing not only cytoplasmic cyto C expression but the corresponding mitochondrial fractions. More importantly, does inhibition of mitochondrial apoptosis (e.g. by overexpressing Bcl2) block DT-NP induced killing.

Response: Thanks for the reviewer's kind suggestion; the western blot analysis of mitochondrial cyto C and Bcl-2 expression was updated in Figure 6d. The results confirmed the down-regulation of Bcl-2 and mitochondrial cyto C.

Figure 6d. Western blotting analysis for the expression of Cyto C, cleaved-caspase 3 and Bcl-2 proteins after treated DT-NPs for 12 h and 24 h. GAPDH was used as an internal control.

(5) The *in vivo* expts. investigating efficacy are impressive (Fig 7), not only in the potent tumour killing (by DT-NP) but the lack of apparent cytotoxicity. With respect to the latter, given the significant accumulation in both liver and kidney I think it is important to investigate parameters of toxicity in these organs.

Response: As suggested, the standard haematology and blood biochemical analysis were performed to examine the potential *in vivo* side effects. There was no detectable abnormal parameters, suggesting favorable safety. Please refer to page 12 in the updated main text. *“Furthermore, the in vivo toxicology and potential side effects were investigated systematically. The standard haematology markers including the white blood cells (WBC), red blood cells (RBC), haemoglobin (HGB), haematocrit (HCT), mean corpuscular volume (MCV), mean corpuscular haemoglobin (MCH), mean corpuscular haemoglobin concentration (MCHC), platelets (PLT), mean platelet volume (MPV) and thrombocytocrit (PCT) were measured (Fig. 9a). Compared with the PBS group, all the parameters in the five treated groups appeared to be normal and the differences between them were not statistically significant (P value > 0.05). These results indicated that these treatments did not cause obvious infection and inflammation in the treated mice⁴⁷.*

Blood biochemical analysis were carried out and various parameters including alanine transaminase (ALT), aspartate transaminase (AST), alkaline phosphatase (ALP), total protein (TP), albumin (ALB), globulin (GLOB), creatinine (CR), blood urea nitrogen (BUN), uric acid (UA) and total bilirubin (TBIL) were examined (Fig. 9b). Compared with the PBS group, no meaningful difference was detected from the five treated groups. Hence, the treatment did not affect the blood chemistry of mice. Furthermore, since alanine transaminase (ALT), aspartate transaminase (AST) and creatinine (CR) are closely related to the functions of the liver and kidney of mice⁴⁷, the results demonstrated that the treatment induced no obvious hepatic and kidney toxicity in mice”.

Figure 9 | *In vivo* side effects evaluation. (a) Haematological data of the mice intravenously injected with different samples at the 21st day post-injection. The terms are noted as followed: white blood cells (WBC), red blood cells (RBC), haemoglobin (HGB), haematocrit (HCT), mean corpuscular volume (MCV), mean corpuscular haemoglobin (MCH), mean corpuscular haemoglobin concentration (MCHC), platelets (PLT), mean platelet volume (MPV) and thrombocytocrit (PCT). (b) Blood biochemical analysis at the 21st day post-injection. The terms are following: alanine transaminase (ALT), aspartate transaminase (AST), alkaline phosphatase (ALP), total protein (TP), albumin (ALB), globulin (GLOB), creatinine (CR), blood urea nitrogen (BUN), uric acid (UA), and total bilirubin (TBIL).

(6) Fig 8/discussion thereof – it is important to note that mitochondrial depolarisation does not equate with mitochondrial permeabilisation (leading to cyt c release) the proposed model should be modified accordingly.

Response: Thanks for the reviewer’s insightful advice. We changed the Depolarization of Mitochondrial transmembrane potential ($\Delta\Psi_m$) into Mitochondrial damage in our description, and previous Figure 8 was updated as Figure 10 in the revised manuscript.

Figure 10. Proposed mechanism for the mitochondria-specific self-circulation of CPT Release and mtROS Burst to damage mitochondria and initiate cell apoptosis/death by dual-targeted polyprodrug nanoreactors.

(7) Other points while readable, the ms. could do with extensive editing to improve its flow/grammar.

Response: As suggested, we polished the manuscript thoroughly.

No related work is in the press with any other journal. No potential conflicts of interest with this work exist. There was no prior discussion with the editors about this work. We trust the scientific merits and the improvement of this work can justify its publication in *Nature Communications*; we look forward to receiving the reviewers’ comments in due course.

Yours sincerely

Prof. Xianglong Hu, Prof. Da Xing
 MOE Key Laboratory of Laser Life Science & Institute of Laser Life Science
 College of Biophotonics, South China Normal University, Guangzhou 510631, P. R. China
 E-mail: xlhu@scnu.edu.cn; xingda@scnu.edu.cn

Reviewers' comments:

Reviewer #1 (Remarks to the Author):

Overall, I am not satisfied at all with the updated manuscript version. Most of my previous concerns remained unanswered. The previous version only discussed H₂O₂; now they tended to include "hydroxyl radical (\cdot OH) and superoxide radical"; may the reviewer also suggest ONOO⁻ and ClO₃⁻ as possibly relevant ROS species?

All data only stays in the level of "qualitative", as stated in the Abstract, for example, "slight free CPT", "initial endogenous mtROS". In most Figures, key experimental details were lacking. In Figure 2d, 1 mM H₂O₂ could trigger the release of ~40% CPT, this is contradictory with previous relevant reports.

In Figures 2d and 2e, the authors describes "1 mg DT-NPs" treated with ROS at different concentrations. The reviewer cannot understand the mass-to-concentration statement. Will the authors treat 1 mg DT-NPs with 1 litre of ROS at different concentrations?.

In the main text, the phrase of "hydroxide radical" has appeared at least four times. The reviewer only knows hydroxide ions.

As stated also by Reviewer 2, most of the data were over-interpreted.

Reviewer #2 (Remarks to the Author):

The reviewers have addressed most of my points satisfactorily. Nevertheless, one outstanding point remains - based on their proposed model mitochondrial dysfunction leads to mitochondrial permeabilisation and apoptosis. To proposd this model, as requested in my original review, Bcl-2 overexpression and/or BAX/BAK deletion (blocking mitochondrial apoptosis) should be used to determine whether this inhibits nanoreactor induced toxicity.

Reviewers' comments:

Reviewer #1:

1. Overall, I am not satisfied at all with the updated manuscript version. Most of my previous concerns remained unanswered. The previous version only discussed H₂O₂; now they tended to include "hydroxyl radical (\cdot OH) and superoxide radical"; may the reviewer also suggest ONOO⁻ and ClO₃⁻ as possibly relevant ROS species?

Response: Thank for the reviewer's insightful comments to improve this work. The reviewer is right! We are sorry to miss the examination of ONOO⁻ and ClO₃⁻ (probable typing error, corrected as ClO⁻), which are both non-radical ROS types with even low endogenous content compared with O₂⁻, H₂O₂, and \cdot OH. As suggested by the reviewer, ONOO⁻ and ClO⁻ were further employed to examine the stimuli-responsive CPT release and nanoparticle degradation. HPLC analysis (Figure 2c) and *in vitro* CPT release (Figure 2d) both demonstrated that DT-NPs could be degraded under these five kinds of ROS, including O₂⁻, H₂O₂, \cdot OH, ClO⁻ and ONOO⁻. Furthermore, obvious degradation and size shrinkage of nanoparticles were also observed by TEM analysis for DT-NPs treated with these five ROS types (Figure 2g). Hence, these five kinds of ROS could mediate the cleavage of thioether linkages and the polydrug degradation to release parent CPT drug.

Please kindly refer to the updated discussion in page 5: "*To investigate the ROS-induced CPT release, KO₂ dissolved into dry DMSO was employed to generate superoxide (O₂⁻),^{5,40} Typical Fenton reaction between Fe²⁺ and H₂O₂ was used to generate hydroxyl radical (\cdot OH),⁴¹ sodium hypochlorite (NaOCl) at pH 6.02 to form hypochlorous acid (HClO), $C_{ClO^-} = Abs_{292nm} / 0.39$ (mM),⁴² ONOO⁻ was prepared according to the reported method, $C_{ONOO^-} = Abs_{302nm} / 1.67$ (mM).⁴³ HPLC analysis showed that PDMA-b-PCPTSM could readily release CPT in the presence of five types of*

ROS at 1 mM, including O_2^- , H_2O_2 , $\cdot OH$, ClO^- and $ONOO^-$ upon 24 h incubation, respectively (Fig. 2c). In vitro CPT release rate was further evaluated for DT-NPs upon treating with these five types of ROS (Fig. 2d). $ONOO^-$ and ClO^- were observed to mediate much faster CPT release compared with O_2^- , H_2O_2 , and $\cdot OH$, which also agreed with their relative oxidation potency.^{4,44}

Please kindly check the discussion in page 6: “After that, the ROS-responsive degradation of DT-NPs was monitored by Transmission Electron Microscope (TEM). For the control group, the diameter of DT-NPs kept to be $\sim 29.3 \pm 5.4$ nm after 24 h incubation in PBS at pH 7.4, 37 °C. Whereas the particle size of DT-NPs decreased readily after 24 h incubation with five types of ROS parallelly at 1 mM; specifically, O_2^- (9.8 ± 2.6 nm), H_2O_2 (9.2 ± 2.6 nm), $\cdot OH$ (8.6 ± 1.9 nm), ClO^- (3.9 ± 1.0 nm) and $ONOO^-$ (3.0 ± 1.2 nm), respectively (Fig. 2g). The degradation extent for the groups of $ONOO^-$ and ClO^- was more significant than other groups, which was expected to promote much faster drug release. Hence, these five kinds of ROS could mediate the cleavage of thioketal linkages and the polyprodrug degradation to release parent CPT drug.”

2. All data only stays in the level of "qualitative", as stated in the Abstract, for example, "slight free CPT", "initial endogenous mtROS". In most Figures, key experimental details were lacking. In Figure 2d, 1 mM H_2O_2 could trigger the release of $\sim 40\%$ CPT, this is contradictory with previous relevant reports.

Response: Thanks for the reviewer’s insightful comments and helpful suggestions, which give us opportunity to improve this work. Please kindly check our further work and following discussion.

(i) As suggested by the reviewer, some language polishing was performed for accurate description, and the experimental details were added accordingly. Furthermore, from a technical point of view, it is hard to real-time monitor the mitochondrial drug release content precisely. Alternatively, based on flow cytometry analysis (FCAS), time-dependent mitochondrial superoxide was evaluated for cancer cells after diverse treatments within 24 h, which was expected to reflect the proposed ROS burst, because mitochondrial superoxide was the source of most ROS types. Please kindly refer the updated discussion in page 9: “Furthermore, the specific mitochondrial superoxide, the source of most ROS species, was also evaluated upon different treatments. FACS and CLSM imaging were employed to discern the superoxide in mitochondria using MitoSOXRed as an indicator.⁵⁷ Real-time kinetic detection of mitochondrial O_2^- by FCAS was performed for living cells upon incubation with diverse samples from 0 h to 24 h, respectively (Fig. 5a and Supplementary Fig. 12). At 10 h, the group of DT-NPs reached the highest peak of mitochondrial O_2^- level, ~ 10.3 -fold higher than the control group, which was much higher than other groups with different targeting properties. Interestingly, upon incubation for 24 h, ~ 6.8 -fold intensity was even found for the group of DT-NPs, suggesting extended highly oxidative stress exerted by the self-circulation of CPT release and ROS burst. Two ROS scavengers, NAC and Vc,

could eliminate ROS in all these groups. Notably, free CPT could also promote moderate elevation of mitochondrial O_2^- at a slightly quick rate, achieving its highest peak at 2 h, ~6.5-fold compared with the control group. Temporarily, fast intracellular diffusion of free CPT with compromised mitochondrial targeting property resulted in the fast upregulation of mitochondrial O_2^- , but the enhancing extent and function duration were lower than that of DT-NPs. After that, the level of mitochondrial O_2^- was evaluated by FACS and CLSM imaging for these groups upon 8 h incubation, highest red fluorescence was observed for the group of DT-NPs. It agreed well with above kinetic monitoring, demonstrating most significant mitochondrial O_2^- generation from DT-NPs, potentially endowed by the in-situ mitochondrial CPT release and ROS burst (Fig. 5b, c, d).”

(ii) There are only a few literatures using ROS-responsive thioketal linkages to covalently conjugate drug for ROS-responsive drug release (typically, Farokhzad et al. *Adv. Mater.* 2017, 29, 1700141; Liu et al. *Angew. Chem. Int. Ed.* 2014, 53, 1-7), even though many drug-free carriers with thioketal groups in the backbone were reported for ROS-sensitive degradation and loaded cargo release (typically, Murthy et al. *Nat. Mater.* 2010, 9, 923-928; Xia et al. *Angew. Chem. Int. Ed.* 2013, 52, 6926-6929). ROS-responsive cleavage of thioketal linkage is well demonstrated in these works. In the work by Prof. Liu et al. (*Angew. Chem. Int. Ed.* 2014, 53, 1-7, page 3, Figure 1d), *in situ* generated ROS from light radiation-induced photodynamic process was used to cleave the thioketal linkage, ~5% cumulative DOX release was found upon incubation for ~4 h (240 min) at dark without light irradiation, the data with long incubation time was not found in the work.

On the other hand, in the work of Prof. Farokhzad et al. (*Adv. Mater.* 2017, 29, 1700141, page 3, Figure 2B), KO_2 (0-100 μ M) was used to simulate ROS-triggered MTO drug release. After incubating with 100 μ M KO_2 for 24 h, ~30% MTO release was observed in Prof. Farokhzad's work, and ~20% CPT release was determined for DT-NPs after 24 h incubation with 100 μ M KO_2 in current manuscript (Figure 2d). The polymer composition and drug moieties were totally different, herein, we inclined to consider that the data were not contradictory with each other in terms of the drug release extent. In addition, H_2O_2 was not examined for drug release in Prof. Farokhzad's work, the oxidation efficacy of KO_2 was stronger than H_2O_2 , but the stability was much lower than H_2O_2 , thus the drug release rate in the presence of KO_2 probably be lower than that of H_2O_2 , which agreed well with the result in Prof. Xia's work, in which the degradation extent in the H_2O_2 group was slightly higher than that of KO_2 group (*Angew. Chem. Int. Ed.* 2013, 52, 6926-6929, Supporting Information, page 11, Figure S5). Similar result was also confirmed by the relative drug release rate (Figure 2d) and the nanoparticle degradation observed by TEM in our work (Figure 2g). The drug release results were readily repeated, indicating ~40% CPT release at 1 mM H_2O_2 (Figure 2d, 2e), which was in reasonable range compared with these published works.

Furthermore, in the work of Prof. Xia et al. (*Angew. Chem. Int. Ed.* 2013, 52, 6926-6929), ROS-responsive thioketal linkages were located in the polymer backbone of poly(amino thioketal)

(PATK). Although high content of H₂O₂ (50 mM, 100 mM, and 200 mM) was employed to study the significant degradation of PATK by ¹H NMR analysis (page 2, Figure 2a), which was partially due to high content necessity of polymers for ¹H NMR analysis compared with ¹H NMR analysis for small molecules. Notably, in the Supporting Information part (page 11, Figure S5 caption), the authors also claimed that: “*The significant destabilization of DNA/PATK polyplexes was observed under biologically relevant levels of H₂O₂ such as 100 μM and 1 mM H₂O₂...*”. It confirmed that the polymer backbone could be cleaved in the presence of 100 μM and 1 mM H₂O₂, which was comparable with the experimental condition and results in our manuscript.

In summary, the ROS-responsive polyprodrug degradation was readily repeatable in this manuscript. Despite the distinct difference of polymer composition and drug moieties, our observed results were reasonably comparable with many published excellent papers.

3. In Figures 2d and 2e, the authors describes "1 mg DT-NPs" treated with ROS at different concentrations. The reviewer cannot understand the mass-to-concentration statement. Will the authors treat 1 mg DT-NPs with 1 litre of ROS at different concentrations?.

Response: Thanks for the reviewer’s insightful comment to point out the confusing description in our manuscript. In previous version, we referred to the work of Prof. Xia *et al.* (*Angew. Chem. Int. Ed.* 2013, 52, 6926-6929, Supporting Information, Page 3, last paragraph): “*A degradation rate of thioketal linkages in PATK under ROS condition was investigated by incubating PATK (5 mg, 10.82 μmol of thioketal groups) with different concentrations of H₂O...*”

As suggested by the reviewer, we added the volume message in the figure captions to further clarify the detail. Please kind refer to updated Figures 2d and 2e: “*(d) In vitro CPT release from DT-NPs against five kinds of ROS for 24 h with diverse contents. (e) Degradation kinetics of DT-NPs, determined by the normalized scattered light intensities from DLS analysis and (f) In vitro CPT release from DT-NPs against different level of H₂O₂ at pH 7.4, 37 °C. (g) TEM images of DT-NPs after 24 h incubation with five types of ROS types at 1 mM, 37 °C, scale bar, 200 nm. The content of DT-NPs (1 mg in 0.5 mL, 0.82 μmol of thioketal groups) was employed in (c)-(g).*”

4. In the main text, the phrase of "hydroxide radical" has appeared at least four times. The reviewer only knows hydroxide ions.

Response: The reviewer is right! We are sorry for the misspelling, it should be “hydroxyl radical”, which has been corrected in the updated version, and we polished the manuscript thoroughly.

5. As stated also by Reviewer 2, most of the data were over-interpreted.

Response: Thanks for the reviewer’s constructive comments and suggestions, which greatly improve the quality of this work. After deliberate work, the proposed ROS burst was demonstrated at diverse levels, including solution state, cellular level, as well as tumor-bearing mice. We

cordially invite the respected reviewer to re-evaluate the manuscript and give valuable comments. Thanks again for the reviewer's kind review and potential positive comments.

Reviewer #2:

The reviewers have addressed most of my points satisfactorily. Nevertheless, one outstanding point remains - based on their proposed model mitochondrial dysfunction leads to mitochondrial permeabilisation and apoptosis. To propose this model, as requested in my original review, Bcl-2 overexpression and/or BAX/BAK deletion (blocking mitochondrial apoptosis) should be used to determine whether this inhibits nanoreactor induced toxicity.

Response: Thanks for the reviewer's kind review and constructive advice. As suggested, western blot analysis was further performed to evaluate the Bcl-2 overexpression and/or BAX/BAK deletion (Figure 6d, Supplementary Figure 13).

Please kindly refer the updated discussion in page 11: *“The apoptosis-related proteins in 4T1 cells were further detected by western blot analysis (Fig. 6d and Supplementary Fig. 13). After incubation with DT-NPs for 12 h or 24 h, the expression of cytochrome c in cytoplasm and caspase-3 both increased dramatically, and the pretreatment with antioxidants (NAC or Vc) would remit the upregulation of cytochrome c and cleaved caspase-3 in cytoplasm. Conversely, the expression of anti-apoptotic Bcl-2 protein was significantly inhibited, and the pro-apoptotic Bax increased greatly. Obviously, these results verified the mitochondria mediated apoptotic pathway triggered by DT-NPs.³⁵ Based on the above results, it could be concluded that mtROS triggered in-situ mitochondrial CPT release could definitely amplified oxidative stress (mtROS) and decrease the mitochondrial membrane potential, which would result in remarkable mitochondrial damage to initiate programmed cell death. Herein, the dual-targeting polyprodrug nanoreactors could act as polyprodrug nanoreactors to endogenously activate in-situ mitochondrial drug release and ROS burst, exerting persistent oxidative stress for enhanced cancer chemodynamic therapy.”*

This work was solely submitted in Nature Communications. No related work is in the press with any other journal. No potential conflicts of interest with this work exist. There was no prior discussion with the editors about this work. We trust the scientific merits and the improvement of this work can justify its publication in *Nature Communications*; we look forward to receiving the reviewers' comments in due course.

Yours sincerely

Prof. Xianglong Hu, Prof. Da Xing

MOE Key Laboratory of Laser Life Science & Institute of Laser Life Science

College of Biophotonics, South China Normal University, Guangzhou 510631, P. R. China

E-mail: xlhu@scnu.edu.cn; xingda@scnu.edu.cn

REVIEWERS' COMMENTS:

Reviewer #3 (Remarks to the Author):

The authors have addressed all the comments of the reviewers in the revised manuscript and I recommend it publish on nature communications after the following spelling mistakes etc. are corrected.

1. The abbreviation 'DT-NPs' is not suitable for 'dual-targeting polyprodrug nanoreactors'. Should it be 'DT-PNs'?

2. Line 66, 'modal' should be 'model'?

3. Line 597, 'c1' should be '1'? The 2 in H₂O₂ should be subscript.

4. In Supplementing Information:

In Supplementary Figure 4 and 5: The English letters labeled for chemical structures are not corresponding with each other in the same figure. They should be corrected as which in Supplementary Figure 2.